# Addressing Pitfalls in the Evaluation of Uncertainty Estimation Methods for Natural Language Generation

**Mykyta Ielanskyi**[1], **Kajetan Schweighofer**[1], **Lukas Aichberger**[1], **Sepp Hochreiter**[1,2]

[1] ELLIS Unit Linz and LIT AI Lab, Institute for Machine Learning,
  Johannes Kepler University Linz, Austria
[2] NXAI GmbH, Linz, Austria
  {ielanskyi, schweighofer, aichberger, hochreit}@ml.jku.at

## Abstract

Hallucinations are a common issue that undermine the reliability of large language models (LLMs). Recent studies have identified a specific subset of hallucinations, known as confabulations, which arise due to predictive uncertainty of LLMs. To detect confabulations, various methods for estimating predictive uncertainty in natural language generation (NLG) have been developed. These methods are typically evaluated by correlating uncertainty estimates with the correctness of generated text, with question-answering (QA) datasets serving as the standard benchmark. However, commonly used approximate correctness functions have substantial disagreement between each other and, consequently, in the ranking of the uncertainty estimation methods. This allows one to inflate the apparent performance of uncertainty estimation methods. We propose using several alternative risk indicators for risk correlation experiments that improve robustness of empirical assessment of uncertainty estimation algorithms for NLG. For QA tasks, we show that marginalizing over multiple LLM-as-a-judge variants leads to reducing the evaluation biases. Furthermore, we explore structured tasks as well as out of distribution and perturbation detection tasks which provide robust and controllable risk indicators. Finally, we propose to use an Elo rating of uncertainty estimation methods to give an objective summarization over extensive evaluation settings.

## 1 Introduction

Predictive uncertainty has been linked to the occurrence of a subset of hallucinations known as confabulations (Farquhar et al., 2024). Such confabulations are sequences generated by a large language model (LLM), that have no support in either the training set of the model nor in the prompt. The expressivity of natural language allows these models to obfuscate their lack of knowledge in a manner that can be challenging to detect. Therefore, uncertainty estimation is essential to detect such confabulations and ensure the reliability and wider applicability of LLM-based systems.

Predictive uncertainty in natural language generation (NLG) can be quantified by the entropy of the LLMs predictive distribution (Malinin & Gales, 2020). In the literature on uncertainty estimation in univariate classification, predictive uncertainty is often decomposed into aleatoric and epistemic components (Gal, 2016). The aleatoric uncertainty can be attributed to the inherent stochasticity of the prediction, while the epistemic uncertainty arises from lack of knowledge of the true model parameters (Schweighofer et al., 2023). In case of confabulation detection in NLG, most of the time the aleatoric uncertainty of predicting with a given model with parameters $w$ for a new input $x$ is considered (Kuhn et al., 2023; Aichberger et al., 2025).

Currently, uncertainty estimation algorithms for NLG are evaluated mostly in terms of selective prediction on a narrow class of problems which is question-answering datasets, see Tab. 1. The motivation for using QA tasks is that the ability to effectively retrieve information could be linked to factuality and hallucinations while having a relatively low demand for model performance. However, this class of problems is characterized by short length of the expected answer and impreciseness of the

Table 1: Evaluation protocols recently used for uncertainty estimation in NLG. Few works evaluate their methods beyond selective prediction on QA tasks and rely on approximate correctness functions or a small number of human correctness evaluations.

| REFERENCE | TASK | CORRECTNESS FUNCTION |
|---|---|---|
| MALININ & GALES (2020) | TRANSLATION | BLEU |
| FOMICHEVA ET AL. (2020) | TRANSLATION | HUMAN |
| KUHN ET AL. (2023) | QA | ROUGE-X |
| FADEEVA ET AL. (2023) | QA, SUMMARIZATION | ROUGE-X, BERTSCORE |
| DUAN ET AL. (2024) | QA | ROUGE-X |
| MANAKUL ET AL. (2023) | FACT VERIFICATION | HUMAN |
| FARQUHAR ET AL. (2024) | QA | JUDGE |
| BAKMAN ET AL. (2024) | QA | JUDGE |
| AICHBERGER ET AL. (2025) | QA | ROUGE-X, BLEURT |
| CHEN ET AL. (2024A) | QA | ROUGE-X |
| KOSSEN ET AL. (2024) | QA | JUDGE, F-1 |
| NIKITIN ET AL. (2024) | QA | JUDGE |
| AICHBERGER ET AL. (2024) | QA | JUDGE, F-1 |
| ABBASI-YADKORI ET AL. (2024) | QA | F-1 |

ground truth solution. Importantly, the evaluation of an answer is done by approximate correctness functions, such as comparing substrings or utilizing text similarity models. These correctness functions have been criticized and are often not considered robust (Schluter, 2017; Zheng et al., 2023), yet are widely used in NLG.

Santilli et al. (2024) concurrently investigate the relation between Rouge-L, LLM-as-a-judge and Human annotators and the impact it has on the empirical performances reported in Farquhar et al. (2024) and Fadeeva et al. (2023). They conclude that LLM-as-a-judge should be preferred as a correctness metric in such assessments, and the effects of thresholds should further be investigated with sequence length being an important factor in variability of outcomes. At the same time, Zheng et al. (2023), the original work proposing LLM-as-a-judge for assessing correctness, already point out biases inherent to the approach. Dorner et al. (2025) further state that even high agreement to human annotators may be insufficient to mitigate the evaluation biases of the judge models. Overall, there remain many open questions regarding current practices in evaluating uncertainty estimation in NLG settings. In our work, our objective is to provide insight into the pitfalls of current practices and recommendations to improve on them.

Specifically, our contributions are as follows:

- We conducted a detailed investigation of weaknesses of the evaluation practices used in recent work on uncertainty estimation in NLG and show that one of their leading causes is the lack of marginalization over the correctness function in selective prediction.
- We suggest several beneficial alternative risk indicators to be used for risk correlation experiments, including an ensemble of LLM-as-a-judge (Zheng et al., 2023) variants for QA, structured tasks with exact correctness functions, OOD detection, and perturbation.
- We propose using an aggregation technique based on Elo rating (Elo, 1978) for comparing the performance of uncertainty estimation methods across different experimental setups to foster a more objective assessment of their utility and provide additional insights.

## 2 PRELIMINARIES

The uncertainty estimation problem in NLG can be formalized as follows: given an input sequence $\boldsymbol{x} = (x_1, ..., x_\tau) \in \mathcal{X}$ and a model with parameters $\boldsymbol{w}$, we want to infer an uncertainty measure $u : \mathcal{X} \times \mathcal{W} \mapsto \mathbb{R}$. Then $\hat{u}(\boldsymbol{x}, \boldsymbol{w}; \boldsymbol{\theta}_u)$ is an algorithm to obtain an estimate of $u(\boldsymbol{x}, \boldsymbol{w})$, where $\boldsymbol{\theta}_u$ is a vector of hyperparameters of the algorithm.

**Uncertainty estimation methods in NLG.** Recent approaches to uncertainty estimation for NLG estimate uncertainty in a variety of ways. The methods can be loosely categorized into three groups:

those using statistics of a set of sequences from the model, those using a single output sequence and those using heuristics. The first group of methods is based on Monte-Carlo (MC) sampling and Bayesian assumptions with regard to the obtained samples. Such methods base their estimators on some notion of spread in the probability space of the predictive distribution of an LLM (Malinin & Gales, 2020; Kuhn et al., 2023; Aichberger et al., 2025; Chen et al., 2024a; Nikitin et al., 2024). A noteworthy variation of this direction consists of methods that attempt to modify the probabilities of sampled sequences based on the semantic importance of individually generated subsequences (Duan et al., 2024; Bakman et al., 2024) to compensate for the potential impact that they make on the correctness of the predicted sequence. The approaches from the second group use properties of a single generated sequence (Ren et al., 2023; Fadeeva et al., 2023; Kossen et al., 2024; Aichberger et al., 2024) to estimate the model's confidence. Other approaches leverage the facilities of the language model itself or a larger one to determine confidence estimate for a given output sequence (Kadavath et al., 2022; Manakul et al., 2023). A detailed description the methods considered in this work can be found in Apx. B.1.

**Risk correlation experiments.** Intuitively, the fundamental question to which $u(\boldsymbol{x}, \boldsymbol{w})$ should help us find the answer is: "What is the risk of making the prediction for a given input sequence $\boldsymbol{x}$ using the model with parameters $\boldsymbol{w}$?". This connection of uncertainty and risk has recently been advocated in the univariate classification setting (Lahlou et al., 2023; Kotelevskii & Panov, 2025). In accordance with this perspective, the utility of uncertainty estimation methods is empirically evaluated as a correlation between the estimated uncertainty $\hat{u}(\cdot)$ and some risk indicator $r(\cdot)$ on sets of predictions, defined as

$$\xi \;=\; Cor\left[(\hat{u}(\boldsymbol{x}_i, \boldsymbol{w}; \boldsymbol{\theta}_u))_{i=1}^N, (r(\boldsymbol{x}_i, \boldsymbol{y}_i'))_{i=1}^N\right] \;. \tag{1}$$

Here, $Cor$ is a correlation metric and $\boldsymbol{y}'$ is the predicted output sequence of the LLM. We do not assume a linear relation between the risk and the uncertainty, which restricts eligible $Cor$ to rank correlation metrics with area under the ROC curve (AUROC) being the most commonly used. Based on the risk indicators used for evaluation of uncertainty estimation algorithms in the univariate classification literature (Welling & Teh, 2011; Gal & Ghahramani, 2016; Lakshminarayanan et al., 2017; Malinin & Gales, 2018; D' Angelo & Fortuin, 2021; Daxberger et al., 2021; Mukhoti et al., 2023; Schweighofer et al., 2023), we can distill the following empirical properties that uncertainty estimate $\hat{u}$ must possess:

1. $\hat{u}$ is higher for $\boldsymbol{x}' \sim \mathcal{D}_{\text{test}}$ than for $\boldsymbol{x} \sim \mathcal{D}_{\text{test}}$ if the risk of prediction using $\boldsymbol{w}$ (aleatoric) or $\boldsymbol{w} \sim p(\boldsymbol{w} \mid \mathcal{D})$ (epistemic) for $\boldsymbol{x}'$ is higher than for $\boldsymbol{x}$.

2. $\hat{u}$ is not lower for $\boldsymbol{x}'$ than for $\boldsymbol{x} \sim \mathcal{D}_{\text{test}}$ if $\boldsymbol{x}'$ is drawn from a different data generating function than one that produced the training data $\mathcal{D}$.

3. $\hat{u}$ is not lower for $\boldsymbol{x}'$ than for $\boldsymbol{x} \sim \mathcal{D}_{\text{test}}$ if $\boldsymbol{x}'$ is obtained from $\boldsymbol{x}$ by some perturbation.

These three empirical properties correspond to the following risk indicators and risk correlation experiments: selective prediction (SP), out-of-distribution (OOD) detection, and perturbation detection. An alternative to risk correlation that is sometimes used for evaluation is an active learning acquisition experiment, which is challenging even in the classification setting (Lüth et al., 2023).

**Selective prediction in NLG uncertainty evaluation.** The current standard for comparing uncertainty estimation methods for NLG is selective prediction on QA datasets (Aichberger et al., 2025; Kuhn et al., 2023; Farquhar et al., 2024; Duan et al., 2024; Bakman et al., 2024). The approximate correctness function $c : \mathcal{Y}' \times \mathcal{Y} \times \mathcal{X} \mapsto \{0, 1\}$ maps the prompt $\boldsymbol{x} \in \mathcal{X}$, provided reference answer $\boldsymbol{y}' \in \mathcal{Y}'$ and a generated answer $\boldsymbol{y}' \in \mathcal{Y}$ to a binary value which indicates the correctness of the generated answer. $c$ can be parametrized by a parameter vector $\boldsymbol{\theta}_c$. The negated correctness $\neg c(\dots)$ or 'incorrectness' is then a risk indicator that is used in the SP experiments as follows:

$$\xi_{\text{SP}} \;=\; Cor\left[(\hat{u}(\boldsymbol{x}_i, \boldsymbol{w}; \boldsymbol{\theta}_u))_{i=1}^N, (\neg c(\boldsymbol{y}_i', \boldsymbol{y}_i, \boldsymbol{x}_i; \boldsymbol{\theta}_c))_{i=1}^N\right] \;. \tag{2}$$

$\xi_{\text{SP}}$ captures the uncertainty scores ability to distinguish between correct and incorrect predictions. It represents the probability that a randomly chosen correct sample is ranked higher than a randomly chosen incorrect sample in terms of the uncertainty score.

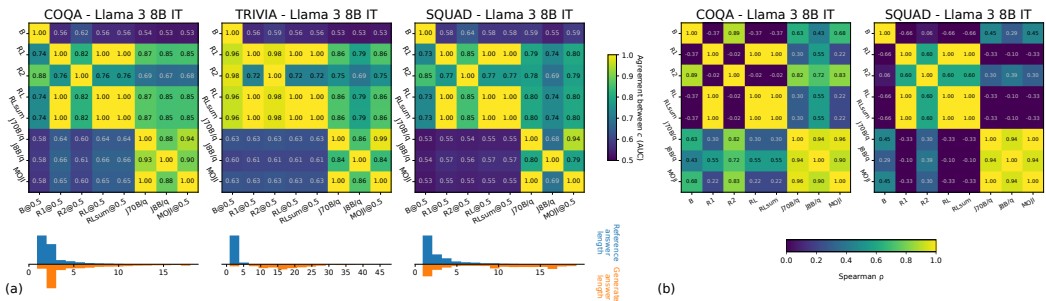

Figure 1: Approximate correctness consistency on selected QA datasets. R indicates ROUGE family, B - BLEU. judge models are indicated with J, 'q' stands for QA prompt used in Farquhar et al. (2024) (see Sec. C.2 for more details on prompting). **(a)** Agreement of correctness metrics in terms of mutual AUROC (not symmetric). Column values are binarized at $0.5$ where applicable. **(b)** Agreement on the ranking of UE algorithms when labeled by the pair of approximate correctness functions. $\rho$ of 1 indicates identical ordering while $\rho$ of 0 indicates uncorrelated rank assignment by the two correctness functions.

## 3 PITFALLS OF THE CURRENT EVALUATION PROTOCOL

In the univariate classification setting, the correctness function is very simple, usually consisting of selecting the highest probability output class and checking its identity to the class provided as the label. In NLG, correctness algorithms are more complex due to the large space of possible sequences and a certain degree of invariance to syntactic permutations and paraphrasis. Selective prediction performance in Eq. (2) depends on both the quality of the uncertainty estimates and the bias and variance of the correctness labels.

**Approximate correctness functions in NLP.** The standard substring matching correctness algorithms are the ROUGE (Lin, 2004) and BLEU families (Papineni et al., 2002). To turn these into correctness functions, one is required to specify a threshold $d$ and the n-gram parameter $n$, so $\boldsymbol{\theta}_c = (d, n)$. Learned correctness functions, such as BERTScore (Zhang et al., 2020) and BLEURT (Sellam et al., 2020) use the similarity of the answer and the reference in an embedding space. LLM-as-a-judge (Zheng et al., 2023) prompts an LLM to confirm the correctness of the answer with respect to the reference. Further reference on correctness functions are provided in Apx. B.4. The key properties these approximate correctness functions have in common is that they are parametric and rely on some notion of similarity of the provided answer to a reference one. The specific set of parameters used has a prominent effect on the labels they produce.

**Effects of bias and variance in correctness labels on AUROC.** Let us consider two scenarios of perturbation of labels when computing sample AUROC for risk correlation experiment. In the first scenario a random, example independent Bernoulli noise is added to the labels. Such noise perturbs the label with a certain probability $p$. In Apx. D.2 we show that such perturbations lead to the following transformation of the original sample AUROC:

$$\text{AUROC}^{\text{noisy}} = \text{AUROC}^{\text{orig}} \cdot (1 - 2p) + p \tag{3}$$

The second scenario that we consider is one where a sample dependent distortion $d_{x_i}$ has been applied to the labels. In case of selective prediction, this distortion would correspond to a bias of the correctness function, where it systematically assigns incorrect labels to specific input-output pairs:

$$c_{x_i}^{\text{biased}} = \begin{cases} c_{x_i} & \text{if } d_{x_i} = 0 \\ \neg c_{x_i} & \text{if } d_{x_i} = 1 \end{cases} \tag{4}$$

Table 2: Adversarially selecting a correctness function on the QA benchmark to improve the ranking of individual uncertainty estimation methods. The values are frequencies of uncertainty estimation methods Top-3 membership on the considered QA datasets. The reference for our assessment is an average over LLM-as-a-judge variants introduced in Sec. 4. Details on the settings used to produce this table can be found in Appx. B.5.3.

| METHOD | REFERENCE | ADVERSARIAL | INCREASE |
|---|---|---|---|
| PREDICTIVE ENTROPY | 0.000 | 0.188 | +0.188 |
| PREDICTIVE ENTROPY (LN) | 0.000 | 0.125 | +0.125 |
| SEQUENCE LENGTH (SAMPLE) | 0.250 | 0.312 | +0.062 |
| SEQUENCE LENGTH (ANSWER) | 0.312 | 0.562 | +0.250 |
| EIGENSCORE | 0.125 | 0.250 | +0.125 |
| TOKENSAR | 0.062 | 0.062 | +0.000 |
| SENTENCESAR | 0.438 | 0.556 | +0.118 |
| SAR | 0.125 | 0.188 | +0.062 |
| PERPLEXITY | 0.125 | 0.444 | +0.319 |
| MIN TOKEN LOG-PROBABILITY | 0.125 | 0.500 | +0.375 |
| SEMANTIC ENTROPY | 0.125 | 0.333 | +0.208 |
| SEMANTIC ENTROPY (LN) | 0.562 | 0.667 | +0.104 |
| P(TRUE) | 0.250 | 0.375 | +0.125 |
| G-NLL | 0.375 | 0.688 | +0.312 |

Furthermore, such a distortion would result in an AUROC that could be decomposed into an interpolation between the AUROC of the undistorted samples and the distorted ones:

$$\text{AUROC}^{\text{dist}} = \text{AUROC}^{\text{orig-undist}} \frac{n_0(d_i = 0) \, n_1(d_j = 0)}{n_0 \, n_1} \tag{5}$$
$$- \text{AUROC}^{\text{orig-dist}} \frac{n_0(d_i = 1) \, n_1(d_j = 1)}{n_0 \, n_1} + 0.5 \left( \frac{n_0(d_i = 1)}{n_0} + \frac{n_1(d_j = 1)}{n_1} \right)$$

Detailed derivation and validity of the estimator per sample size of this identity can be found in Apx. D.2. Independent Bernoulli noise on the labels affects all UE methods equally in the asymptotic case (Eq. (3)). At the same time, bias in correctness estimates affects the ranking proportionally to the a) proportion of distorted samples; b) the discrepancy in ranking quality on the distorted and undistorted samples. Therefore, if we do not marginalize the random noise in the risk indicator, it will have the effect of a sample-dependent distortion, which can affect the apparent performances of different methods differently. *Most prior work ignores the non-deterministic nature of the correctness estimates used.* This is particularly relevant for the LLM-as-a-judge approach, since it entails multiple sources of stochasticity to obtain the correctness label. The correctness label may change upon sampling again without any further changes, different prompts or between model families.

**Disagreement of different correctness functions in QA.** In Fig. 1 (a) we compare the predictions of widely used correctness functions on the QA datasets commonly used for comparing NLG uncertainty estimation algorithms. We observe that the n-gram based correctness function families BLEU and ROUGE show substantial disagreement between each other and the LLM-as-a-judge. Similar observations have been independently reported by Santilli et al. (2025). Different variants of ROUGE show high agreement among them in some scenarios. This agreement can be largely explained by short reference answers provided for the QA datasets, rendering these n-gram based metrics equivalent in most scenarios. As can be seen in the bottom part of Fig. 1 (a), the reference answer lengths are very short, with most consisting of only one or two words. This demonstrates bias and noise in approximate correctness estimators. Furthermore, during our investigation, we have discovered an artifact in a widely used ROUGE-2 and BLEU implementation (Luong et al., 2017) described in more detail in Apx. B.4.1.

**Inconsistency of uncertainty estimation method ranking.** Fig. 1 (b) depicts Spearman correlations between the ranks of NLG uncertainty estimation algorithms evaluated on the given datasets according to different correctness functions. On both CoQA and SQuAD it can be observed that the disagreement in ranking uncertainty estimation methods falls on the lines between judge and n-gram

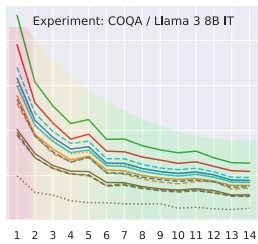
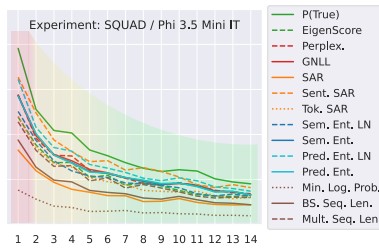

Figure 2: Bootstrap estimate of the standard deviation of mean of AUC performance on selected QA dataset / model combinations. As a rule of thumb, using SP-MoJI with 4 judges reduces the standard deviation of performance estimator twofold. For implementation details refer to B.5.1.

methods with a noticeable BLEU / ROUGE-2 artifact. The adaptive ROUGE metrics are in perfect agreement with ROUGE-1 due to the low length of the reference answers. The judge models agree more with the BLEU / ROUGE-2 than with other ROUGE variants. This indicates, that among the approximate methods the LLM-as-a-judge might be the more reliable one, although not universally.

**Correctness-hacking of QA benchmarks.** In Tab. 2 we show results of optimizing the performance of uncertainty estimation methods with respect to the correctness function. The experiment shows that the apparent performance of the methods can often be improved substantially compared to the value obtained for $\tilde{c}_{\text{reference}}$ by selecting an opportune correctness function $\tilde{c}$ and parametrization $\theta_c$. This also holds for some of the introduced heuristic uncertainty measures, like the sequence length. This further highlights the vulnerabilities of the current evaluation strategy.

## 4 IMPROVING THE EVALUATION WITH ROBUST RISK INDICATORS

In this section, we propose remedies to the issues described in Sec. 3 by introducing several reliable risk indicators, including ones inspired by the univariate classification setting as described in Sec. 2.

**Exact correctness.** If we use problems with deterministic and non-parametric correctness function $c_e$, we can eliminate the need for marginalizing the correctness label. In this case $c := c(\boldsymbol{y}_i, \boldsymbol{x}_i)$ has no parameters that need marginalization. We refer to this deterministic non-parametric correctness function as *exact correctness*. Practical tasks where exact correctness is defined would be problems that are non-trivial to solve but can be verified symbolically. Some examples are constrained text generation, code generation and mathematical problems. These are often called *structured problems*.

**Marginalizing the variability of approximate correctness.** Judge models are subject to biases and uncertainty with respect to sampling (Zheng et al., 2023) even when they show high agreement to human labels Sicilia et al. (2024); Dorner et al. (2025). Moreover, the output is sampled from the judge models stochastically. We propose using Selective Prediction using Mixture of Judges and Instructions (SP-MoJI) as a method for evaluating performance of NLG UE methods on datasets where approximate correctness usage is unavoidable. With SP-MoJI we make multiple judge LM invocations, compute the $\xi$ for each one of them (the computed uncertainties stay the same) and take a mean in order to marginalize over the parameters of correctness label in Eq. (2). This reduces UE evaluation biases due to judge model, sampling, prompt and model family:

$$\xi_{\text{SP-MoJI}} = \mathbb{E}_{\theta_c}[\xi_{\text{SP-J}}] \approx \frac{1}{K}\sum_{k=1}^{K} Cor\left[(\hat{u}(\boldsymbol{x}_i, \boldsymbol{w}; \boldsymbol{\theta}_u))_{i=1}^{N}, (\neg J_k(\boldsymbol{y}_i', \boldsymbol{y}_i, \boldsymbol{x}_i; \boldsymbol{\theta}_k))_{i=1}^{N}\right]. \quad (6)$$

We refer to the average of LLM-as-a-judge prediction when used as a correctness function as MoJI (without SP). Note that simply plugging MoJI correctness into Eq. (2) would not be algebraically equivalent to SP-MoJI in Eq. (6) - the first is an inner expectation while SP-MoJI is an outer expectation. At the same time the entropy of MoJI label can be used for excluding problematic QA entries from evaluation (Apx. E.1). Such marginalization accounts for the aleatoric and epistemic uncertainty of the judge model.

To further motivate SP-MoJI we have analyzed the spread of performance estimates depending on the number of judges used in Fig. 2. Using bootstrapping, we computed the standard deviation of the estimator of UE method performance using different number of diverse judge models. When using a single judge the SD can reach $0.04$, which is roughly equivalent to $\pm 8\%$ confidence interval at 95% confidence. This is on the order of magnitude of differences between methods in e.g. Tab. 2. When using more judge evaluations with different prompts and models the SD of the performance estimates reduces considerably. At the same time, we observe that the benefits of additional judge calls diminish past about 10 invocations. Small scale manual annotation experiment shows that MoJI has better agreement with humans than the individual judges have on average (Appx. B.5.4).

**OOD label and perturbation as risk indicators.** In the univariate classification setting, OOD detection and perturbation detection tasks are considered alongside selective prediction (Apx. D.1). In these tasks it is assumed, that the risk of using the model on an input that violates the i.i.d. assumption or is corrupted is higher than that of an in distribution example. The evaluation would then use the OOD identifier $o : \mathcal{X} \mapsto \{0, 1\}$ as a risk indicator. Unlike the image domain, obtaining OOD examples for text data is difficult. When thinking of OOD examples for text, one would imagine questions about things that have not yet come to be or are otherwise unknown or ambiguous in the general text corpora. The OOD datasets, therefore, need to be constructed artificially.

Another alternative is perturbation detection. The evaluation objective then takes a slightly different form. Let $p$ be a corruption function that perturbs input $\boldsymbol{x}_i$ with strength $s_p$, then perturbation detection objective in accordance with Prop.3 in Sec. 2 is as follows:

$$\xi_{\text{perturb}} = \frac{1}{N} \sum_{i=1}^{N} Cor\left[\hat{u}(p(\boldsymbol{x}_i, s_p), \boldsymbol{w}; \boldsymbol{\theta}_u), s_p\right] . \tag{7}$$

**Empirical implementation of the proposed risk indicators.** As structured tasks we select *code completion* and *constrained text generation*. Code completion problems allow for non-parametric correctness verification by means of unit tests. We picked BigCodeBench (BCB) (Zhuo et al., 2024) due to its convenience and the fact, that it focuses on applied python problems which are well represented in pretraining sets of language models. As a constrained text generation problem we selected COLLIE (Yao et al., 2024) dataset which provides non-parametric evaluation pipeline.

Several datasets seek to provide artificial OOD examples. Known-Unknowns (Amayuelas et al., 2024) seek to collect questions that can be assumed to have controversial answers in the common training sets. SQuADv2 (Rajpurkar et al., 2018) provides questions formulated to be unanswerable given the prompt. For perturbation detection we perturb two QA datasets: CoQA and SQUADv2. We obtain them by randomly shuffling the words in the story to which the corresponding question is related. We control the perturbation strength by setting the proportion of words that are displaced.

For the lack of better reference, we evaluated MoJI as a correctness function on selected structured tasks. Apx.Fig. 10 (a) shows the agreement between the correctness metrics. Panel (b) shows the Spearman correlation between ordering of UE methods. The approximate correctness functions struggle to match the exact ranking on COLLIE, with only the largest models with specifically modified prompt being correlated to exact correctness. This is due to the fact, that the reference sequence may have completely different semantics compared to the one generated by the LLM, while both fulfilling the specified requirements. Individual judges fail to pick this up unless the prompt is adapted to the structured tasks and shows high discrepancy between the two considered prompts.

## 5 AGGREGATING RESULTS

Once a reliable risk indicator is selected, we are presented with quantitative assessments of uncertainty estimation methods for each model, dataset and sampling parameter considered. It is commonplace to see large tables listing every feasible combination of aforementioned factors which often feature contradicting assessments depending on e.g. the model or datasets used. Depending on how different results are highlighted or how these are discussed in the text, different conclusions, sometimes conflicting, can be drawn from the same raw results. Except Vashurin et al. (2025) that use the average rank, we are not aware of any work that seeks to summarize the available experimental information from testing under diverse datasets and models into a single scalar in a grounded fashion.

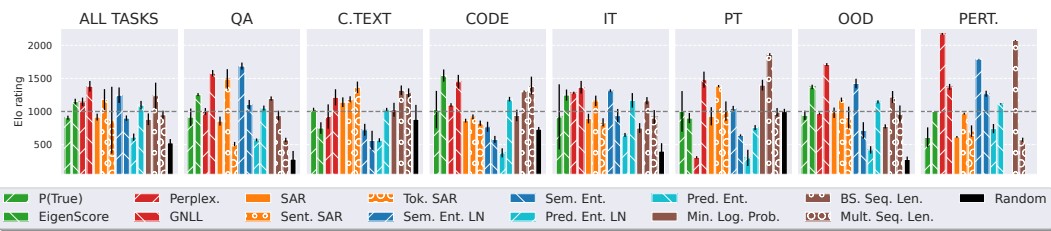

Figure 3: Elo ratings of NLG uncertainty estimation methods. The methods are grouped by color according to their category (see Apx. B.1). The line at 1000 Elo indicates the average rating. Elo rating were independently estimated for several key partitions. Per task used: *QA* - selective prediction on QA datasets, *C.TEXT* - constrained text generation, *CODE* - code completion. Per models used: *IT* - instruction fine tuned models only, *PT* - pretrained models only. Finally, we report the partitions of the alternative risk indicators: *OOD* - out-of-distribution and *PERT* - perturbation.

**Elo rating of uncertainty estimation methods.** Drawing inspiration from popular approaches used for general evaluation of LLM skills (Chiang et al., 2024), we use the Elo rating system (Elo, 1978) to gain high level insight into performance of the considered uncertainty estimation methods. Originally intended to rate skill of chess players, the Elo rating provides an iterative algorithm to compute relative performance of players based on pairwise comparisons (games). We will treat each independent dataset / model risk correlation experiment as a separate game, where the players, methods A and B, can win the game by having higher performance according to Eq. (1). The pairs and experimental runs are then sampled uniformly until the ratings converge to a stationary distribution that is defined by their relative per problem performance (Cortez & Tossounian, 2024). While each prediction in each considered dataset could be considered a separate game when estimating the Elo ratings, for the sake of consistency and to avoid unnecessary additional complexity we only consider the outcomes of experiments on the full datasets.

One advantage of the Elo system over rank averaging is the probabilistic interpretability of the scores. With the usual initialization, 400 point difference roughly corresponds to 1 : 10 chances of one method being better than another for a model / dataset combination. In case of rank aggregation as e.g. used in Vashurin et al. (2025), such quantitative information of "how much better?" is not attainable. Another advantage is that it enables for indirect comparisons. For example, if UE methods are evaluated on only partially overlapping sets of tasks, we could retain their relative performances. In Appx. B.5.5 we show that Elo score handles an artificially induced mild version of such scenario better than rank aggregation. Finally, the Elo score naturally accommodates the variability of outcomes within the same experiment and allows prioritizing specific subsets of experiments. This means that we could bootstrap the subsamples of individual experiments to account for the performance variance within a single model - dataset - sampling parameters combination.

The Elo ratings for our experimental suite are presented in Fig. 3. The random baseline allows gauging the relative difficulty of each partition. The *ALL TASKS* ratings are the summary rating of the overall performance of the UE methods considered. The detailed results show that the characteristics required to excel in different partitions of the experimental suite vary. Employing an effective aggregation approach allows us to gain new insights into comparative performance of the UE methods as well as to confirm some of the side note observations made in prior work.

# 6 DISCUSSION & FUTURE WORK

We have shown the theoretical rationale for using SP-MoJI and structured tasks for correlation experiments. While directly assessing improvements to an evaluation protocol is conceptually challenging, we have observed indirect evidence of bias in the existing indicators and reduction thereof through our proposed risk indicators. Furthermore, we have shown that correctness hacking can exploit the bias in order to misrepresent the strengths and weaknesses of UE methods in NLG. We have observed that SP-MoJI is relatively robust to invariances in the selected structured tasks. In Apx. E.1 we further manually inspect a subset of examples with the highest MoJI entropy to assess the quality of the corresponding ground-truth answers.

Upon applying our methodology, we confirm many of the side-note conclusions made in prior work and gain some additional insights. We observe that adjusting the likelihoods of individual tokens based on semantic importance measures appears to be counterproductive in most settings. Similar to recent observations in prior work (Aichberger et al., 2024), we also find that length normalization is harmful in practically all considered scenarios aside for perturbation detection. This could also explain the improved performance of discrete semantic entropy (Farquhar et al., 2024), where the sequence likelihoods are ignored entirely and the entropy is computed based on the semantic cluster size. P(True) does not excel in any of the tasks in particular while EigenScore together with G-NLL appear to perform well for longer sequences in code generation.

The results for perturbation detection are noticeably different from the other tasks considered. Contrary to all other settings, length normalization for both Perplexity and Semantic Entropy appear beneficial. However, the sequence length of the answers generated by the beam search decoding is very strong. Investigating the underlying causes of those findings is an important direction for future work, as well as analyzing possible biases to specific perturbation type and strength.

All methods perform poorly with pretrained models (*PT*), as is evident through the high Elo of the random control baseline on that partition. Perplexity got rated above average on the *IT* partition, which might be a somewhat more realistic assessment than *ALL TASKS*, but there is no specific task in which it could act as the go to choice. This could be related to the lack of structure in the base model outputs and indicates that base models should be treated separately in evaluation. This is relevant to assessment of calibration of base and instruct models when computing the bin statistics.

A key observation is that simple methods have competitive performance in many settings, especially outside the QA domain. This expands on observations of Santilli et al. (2025), as we now state that the performance of the methods not only depends on the underlying correctness, but also on the problem type used for evaluation. It appears that there is no one-size-fits-all in uncertainty estimation for NLG, with different tasks having different UE method preferences. Reduced variability of the risk indicators and aggregation of the results allow us to confidently draw such conclusion.

More recent work on NLG focuses on more advanced modes of language generation such as CoT (Wei et al., 2022) or multi-agent (Chen et al., 2024b) settings. Whereas the setting investigated in this work can be treated as information retrieval from models weights or prompt, these recent modes of NLG can involve much longer generation lengths, iterative decoding processes, multiple models and routing algorithms, etc. These factors further challenge the robustness of uncertainty estimation algorithms. While we leave the empirical analysis of models in these settings to future work, we believe that the basic principle of marginalization over risk function's parameters and avoiding experimental clutter through effective aggregation would be necessary for evaluation in these settings.

## 7 RELATED WORK

Santilli et al. (2025) concurrently investigate the pitfalls of the NLG UE algorithm evaluation employing detailed correlation studies and human annotations. While their analysis of the problem is similar to ours, our work additionally proposes several solutions to the identified pitfalls as well as considers evaluation instability centered around general risk correlation experiments. Recent work by Fadeeva et al. (2023) and Vashurin et al. (2025) provide a comprehensive benchmark suite for UE methods in NLG, utilizing QA, summarization and translation tasks. These two works, however, do not investigate the failure modes of the prevailing evaluation protocol, which is the focus of our work.

Another line of research deals with OOD detection for benchmarking uncertainty estimation in NLG. Vazhentsev et al. (2023) empirically investigate OOD detection on translation, summarization and question answering tasks, comparing density based methods to deep ensembles. Zablotskaia et al. (2023) evaluate the calibration of predictive distributions of LLMs on summarization tasks.

Several works investigate debiasing and uncertainty estimation for LLM-as-a-judge. Liu et al. (2024) introduce a Meta Ranking approach to improve the performance of judge LLMs. Wagner et al. (2024) proposes a method for assessing uncertainty of LLM-as-a-judge for multiple choice answers. Dorner et al. (2025) investigate judge LM bias in general purpose LM evaluation.

## 8 CONCLUSION

In this work we sought to analyze and address pitfalls in the evaluation of NLG uncertainty estimation algorithms. We first formulated a perspective on empirical properties of uncertainty estimation algorithms and the ways they are evaluated. This perspective builds upon the established work in uncertainty estimation for the univariate classification setting, transferred to the natural language generation setting. We further investigate the peculiarities of risk correlation experiments in the NLG UE literature, diagnose the arising issues and propose credible remedies. Overall, our insights and proposed risk indicators aim to foster better evaluation practices and guide the field to further improve UE methods for NLG.

### ACKNOWLEDGMENTS

The ELLIS Unit Linz, the LIT AI Lab, the Institute for Machine Learning, are supported by the Federal State Upper Austria. We thank the projects FWF AIRI FG 9-N (10.55776/FG9), AI4GreenHeatingGrids (FFG- 899943), Stars4Waters (HORIZON-CL6-2021-CLIMATE-01-01), FWF Bilateral Artificial Intelligence (10.55776/COE12). We thank NXAI GmbH, Audi AG, Silicon Austria Labs (SAL), Merck Healthcare KGaA, GLS (Univ. Waterloo), TÜV Holding GmbH, Software Competence Center Hagenberg GmbH, dSPACE GmbH, TRUMPF SE + Co. KG.

### REPRODUCIBILITY STATEMENT

The authors take appropriate measures to ensure reproducibility of the outcomes. Attached to this submission is supplementary material containing (a) complete code used for experiments; (b) intermediate results csv tables, containing every $\xi$ for every dataset, model, uncertainty and correctness measure combination considered. Identities concerning variances in Sec. 3 are derived step by step in Appx.Sec.D along with discussion on about the assumptions and empirical verification.

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

CONTENTS

## A   GENERATIVE AI USAGE DISCLOSURE

The development of this paper saw very limited use of generative AI, which was exclusively used for editing and refining the prose of small portions of human written text.

## B   DATASETS AND METHODS USED IN OUR ANALYSIS

### B.1   CONSIDERED UNCERTAINTY ESTIMATION METHODS

In the following section, we give an overview about the considered uncertainty estimation methods. Two main categories are methods that operate on multiple and single generated output sequences.

Furthermore, an inherent problem of generating output sequences of arbitrary size (though in practice often capped as a maximum length), introduces the problem of having an uncertainty estimate that is independent of the sequence length. For methods based on output probabilities $p(y_t \mid \boldsymbol{x}, \boldsymbol{y}_{<t}, \boldsymbol{w})$, this usually involves non-uniform weighting of individual token probabilities. Finally, we present a set of well performing heuristics.

### B.1.1 MULTIPLE OUTPUT SEQUENCES

Many works investigate measures of sequence-level uncertainty that are defined as expectations over the sequence probability distribution $p(\boldsymbol{y} \mid \boldsymbol{x}, \boldsymbol{w})$ under a given model. Monte-Carlo (MC) approximations thereof rely on sampling multiple output sequences.

**Predictive Entropy.** Similar to the univariate classification setting, Predictive Entropy (Malinin & Gales, 2020) captures the variability in possible outcome sequences. If PE is high, the language model is likely to generate different outcome sequences. However, as the language model does not provide the full predictive distribution $p(\boldsymbol{y} \mid \boldsymbol{x}, \boldsymbol{w})$, but only the conditional distribution $p(y_t \mid \boldsymbol{x}, \boldsymbol{y}_{<t}, \boldsymbol{w})$ for each token. Therefore, estimating the Predictive Entropy necessitates a MC approximation:

$$\mathrm{H}(p(\boldsymbol{y} \mid \boldsymbol{x}, \boldsymbol{w})) = \mathrm{E}_{p(\boldsymbol{y}\mid\boldsymbol{x},\boldsymbol{w})}\left[-\log p(\boldsymbol{y} \mid \boldsymbol{x}, \boldsymbol{w})\right] \tag{8}$$

$$\approx \frac{1}{N}\sum_{n=1}^{N} -\log p(\boldsymbol{y}^n \mid \boldsymbol{x}, \boldsymbol{w}), \qquad \boldsymbol{y}^n \sim p(\boldsymbol{y} \mid \boldsymbol{x}, \boldsymbol{w}).$$

**Semantic Entropy.** Predictive Entropy does not account for the fact that output sequences $\boldsymbol{y}$ are different, yet covey the same semantics. For example, "John is my brother." and "My brother is John" is semantically equivalent, yet are different output sequences. To that end, Kuhn et al. (2023); Farquhar et al. (2024) introduce Semantic Entropy, which accounts for those semantic equivalences. They do so by introducing a semantic cluster probability $p(c \mid \boldsymbol{x}, \boldsymbol{w})$, that is marginalized over possible output sequences:

$$p(c \mid \boldsymbol{x}, \boldsymbol{w}) = \sum_{\mathcal{Y}} p(c \mid \boldsymbol{y}, \boldsymbol{x}, \boldsymbol{w})\, p(\boldsymbol{y} \mid \boldsymbol{x}, \boldsymbol{w}) \tag{9}$$

In practice, Kuhn et al. (2023); Farquhar et al. (2024) suggest to deterministically assign output sequences to clusters. Semantic Entropy (Kuhn et al., 2023; Farquhar et al., 2024) is then defined on this cluster probability distribution:

$$\mathrm{H}(p(c \mid \boldsymbol{x}, \boldsymbol{w})) = \mathrm{E}_{p(c\mid\boldsymbol{x},\boldsymbol{w})}\left[-\log p(c \mid \boldsymbol{x}, \boldsymbol{w})\right] \tag{10}$$

$$\approx \frac{1}{N}\sum_{n=1}^{N} -\log p(c^n \mid \boldsymbol{x}, \boldsymbol{w}), \qquad c^n \sim p(c \mid \boldsymbol{x}, \boldsymbol{w}).$$

Note that while the MC estimate in Eq. (10) is possible if one has access to the cluster probability distribution, this is not the case in practice. Therefore, we use the implementation of Aichberger et al. (2025), who discuss how to construct a proper MC estimator of Semantic Entropy.

**SentenceSAR.** Instead of clustering, Duan et al. (2024) propose to add a consistency dependent penalty to the uncertainty calculation. The resulting measure, SentenceSAR is defined as

$$\text{SentenceSAR} = \frac{1}{N}\sum_{n=1}^{N} -\log p(\boldsymbol{y}^n \mid \boldsymbol{x}, \boldsymbol{w}) + \frac{\sum_{k\neq n} sim(\boldsymbol{y}^n, \boldsymbol{y}^k)\, p(\boldsymbol{y}^k \mid \boldsymbol{x}, \boldsymbol{w})}{\tau}, \tag{11}$$

where $sim(\cdot, \cdot)$ is a semantic similarity BERT-style model and $\tau$ is a temperature parameter. When output sequences $\boldsymbol{y}^n$ are sampled according to the posterior, the left term of Eq. (11) is equivalent to Predictive Entropy. The right term of Eq. (11) can be interpreted as penalty that decreases uncertainty if there are many semantically similar answers. Therefore, SentenceSAR has a similar goal as Semantic Entropy, yet is more or less motivated heuristically.

The SAR method proposed in Duan et al. (2024) combines both SentenceSAR (Eq. (11)) and TokenSAR (Eq. (17)). We consider both SentenceSAR, TokenSAR and SAR in our experiments.

**EigenScore.** The EigenScore method proposed by Chen et al. (2024a) operates in the latent space instead of output probabilities. Due to that, the method aims to better capture semantic information for an accurate assessment of an LLMs likelihood to hallucinate / confabulate. The EigenScore metric is defined as

$$\text{EigenScore} \;=\; \frac{1}{N} \log \det(\boldsymbol{\Sigma} + \alpha \, \boldsymbol{I}_N) \;=\; \frac{1}{N} \log(\prod_{k=1}^{N} \lambda_k) \;=\; \frac{1}{N} \sum_{k=1}^{N} \log(\lambda_k) \,, \qquad (12)$$

where $\boldsymbol{\Sigma} = \boldsymbol{Z}^{\mathsf{T}} \cdot \boldsymbol{J}_d \cdot \boldsymbol{Z}$ is the covariance matrix, $\boldsymbol{Z}$ is a matrix of $N$ sentence embeddings, taken from the latent space of the LLM with dimensionality $d$, $\boldsymbol{J}_d = \boldsymbol{I}_d - \frac{1}{d}\boldsymbol{1}_d$ is the centering matrix where $\boldsymbol{I}_d$ is an identity matrix and $\boldsymbol{1}_d$ is an all-one square matrix of size $d \times d$. The regularization term $\alpha \boldsymbol{I}_N$ with small constant $\alpha$ is added such that $\boldsymbol{\Sigma}$ has full rank. The set $\{\lambda_1, \lambda_2, ..., \Lambda_N\}$ denotes the eigenvalues of the regularized covariance matrix $\boldsymbol{\Sigma} + \alpha \, \boldsymbol{I}_N$, obtained through singular value decomposition. We followed the implementational details by the original authors Chen et al. (2024a). Noteworthy, according to **Remark 1** in Chen et al. (2024a), EigenScore is an approximation of differential entropy in the sequence embeddings space. It can thus be interpreted as a variant of Semantic Entropy, yet not computed in the output space, but in the embedding space. While Semantic Entropy and other methods operating in the output space hinge on the quality of the semantic similarity operation in the output space, EigenScore depends critically on the quality of the sentence embedding space of the LLM.

### B.1.2 SINGLE OUTPUT SEQUENCE

In addition to measures of predictive uncertainty defined as expectations over the sequence probability, also other methods that only consider a single output sequence have been proposed.

**Maximum Sequence Probability.** Similar to the univariate classification setting, the Maximum Sequence Probability has been considered as a measure of uncertainty (Fadeeva et al., 2023). For numerical stability, the negative logarithm of the sequence probability is considered. Formally, the Maximum Sequence Probability (i.e. the negative logarithm thereof) is given by

$$\text{MSP} \;=\; - \max_{\boldsymbol{y}} \log p(\boldsymbol{y} \mid \boldsymbol{x}, \boldsymbol{w}) \,. \qquad (13)$$

Recently, Aichberger et al. (2024) has shown that Eq. (13) is a theoretically justified measure of uncertainty. Approximating Eq. (13) is similarly hard as for other measures of uncertainty in practice, as the autoregressive nature of LLMs makes it necessary to search for the most likely sequence. However, Aichberger et al. (2024) show that the greedily decoded sequence leads to a very efficient estimate that performs very well in practice called `G-NLL`, which is defined as

$$\text{G-NLL} = - \sum_{t=1}^{T} \log \left( \max_{y_t} p(y_t \mid \boldsymbol{x}, \boldsymbol{y}_{<t}, \boldsymbol{w}) \right) \approx \text{MSP} \,. \qquad (14)$$

**Perplexity.** Closely related, the perplexity of an output sequence has been considered as measure of uncertainty (Ren et al., 2023). Note that this is essentially length-normalization as given in Eq. (16), with opposite sign. The perplexity of a sequence $\boldsymbol{y}$ is given by

$$\text{PP} \;=\; \exp \left\{ \frac{1}{T} \sum_{t=1}^{T} - \log p(y_t \mid \boldsymbol{x}, \boldsymbol{y}_{<t}, \boldsymbol{w}) \right\} \,. \qquad (15)$$

### B.1.3 WEIGHTING TOKEN PROBABILITIES

A fundamental problem of calculating uncertainty measures on a sequence basis instead of a token basis is, that there is a depenency on the sequence length $T$. Therefore, short answers are automatically less uncertain than long answers. An ad-hoc solution that is widely regarded in the literature is to use length-normalization (see e.g. Cover & Thomas (2006)). Furthermore, alternatives to this indiscriminative normalization have been proposed, e.g. TokenSAR where individual tokens are weighted according to their semantic relevance (Duan et al., 2024).

**Length-normalization.** Malinin & Gales (2020) popularized the use of length-normalization to make Predictive Entropy comparable across sequence lengths. Instead of the usual sequence probability, the heuristic length-normalized probability distribution

$$\bar{p}(\boldsymbol{y} \mid \boldsymbol{x}, \boldsymbol{w}) \;=\; \prod_{t=1}^{T} p(y_t \mid \boldsymbol{x}, \boldsymbol{y}_{<t}, \boldsymbol{w})^{\frac{1}{T}} \;=\; \exp\left\{\frac{1}{T} \sum_{t=1}^{T} \log p(y_t \mid \boldsymbol{x}, \boldsymbol{y}_{<t}, \boldsymbol{w})\right\} \qquad (16)$$

is considered. Note that this distribution is therefore unnormalized in the sense that the sum over all sequences does not sum up to one. This heuristic has been widely used together with Predictive Entropy, Semantic Entropy or the Maximum Sequence Probability. Using $\bar{p}(\boldsymbol{y} \mid \boldsymbol{x}, \boldsymbol{w})$ instead of $p(\boldsymbol{y} \mid \boldsymbol{x}, \boldsymbol{w})$ in their definitions essentially leads to an additional factor $\frac{1}{T}$ in the definitions of these uncertainty measures. Furthermore, we note that Perplexity is essentially the negative length-normalized sequence probability of an output sequence.

**TokenSAR.** In order to make uncertainty scores comparable across different sequence lengths, instead of summing up token log-likelihoods, one can calculate a weighted average. While length-normalization uniformly weights with one divided by the sequence length, the TokenSAR method by Duan et al. (2024) introduces a weighting dependent on input / output pair $\boldsymbol{x}, \boldsymbol{y}$. The TokenSAR score is given by

$$\text{TokenSAR} \;=\; \sum_{t=1}^{T} -\log p(y_t \mid \boldsymbol{x}, \boldsymbol{y}_{<t}, \boldsymbol{w}) \, \frac{R(y_t, \boldsymbol{y}, \boldsymbol{x})}{\sum_{t=1}^{T} R(y_t, \boldsymbol{y}, \boldsymbol{x})} \qquad (17)$$

with $R(y_t, \boldsymbol{y}, \boldsymbol{x}) = 1 - |sim(\boldsymbol{x} \circ \boldsymbol{y}, \boldsymbol{x} \circ \boldsymbol{y} \backslash \{y_t\})|$. The semantic similarity metric $sim(\cdot, \cdot)$ is a BERT-style model and $\circ$ denotes concatenating two token sequences. Essentially, the weighting term $R$ captures the semantic similarity of an $\boldsymbol{x}, \boldsymbol{y}$ pair and itself, yet leaving out one token of the output sequence. If the similarity chances substantially when removing one token, this one is weighted higher in the weighted average.

The SAR method proposed in Duan et al. (2024) combines both SentenceSAR (Eq. (11)) and TokenSAR (Eq. (17)). We consider both SentenceSAR, TokenSAR and SAR in our experiments.

## B.2 HEURISTICS

Furthermore, we consider popular heuristic methods, that are not grounded in information-theory.

**p(True).** Kadavath et al. (2022) introduced the p(True) baseline to assess the confidence of the model in its own response. The model first generates an answer to a question and then evaluates the probability p(True) — the likelihood that the answer is correct. This is done by prompting the model to assess its own output, such as asking whether the answer is "True" or "False", and using the probabilities assigned to these responses as a confidence score.

**Length of generated answer.** Another heuristic baseline is to consider the length of the generated answers. The reasoning behind this is, that if the model does not know an answer, it will generate longer and more meaningless content as is often observed in public debates. We are not aware of any prior work that has considered it as an uncertainty estimation heuristic, although sequence length plays a role in the analysis in Santilli et al. (2025). The sequences length can be viewed as one of the two components of the G-NLL (Aichberger et al., 2024), since G-NLL can be expressed as the product of sequence length and mean token-level entropy.

## B.3 DETAILS ON THE DATASETS USED

**QA Datasets.** We used several QA datasets CoQA (Reddy et al., 2019), TriviaQA (Joshi et al., 2017) and SQuADv2 (Rajpurkar et al., 2018) that are commonly used in NLG UE literature. SQUADv2 was also used as a dataset with OOD label risk indicator, since it features questions that are intentionally designed to be unanswerable given the prompt. We further used KUQ (Amayuelas et al., 2024) for OOD experiments only as the questions in this dataset are designed to be unanswerable.

**On usage of Truthful QA for UQ evaluation.** We have performed rollouts for Truthful QA (Lin et al., 2021) and evaluated the uncertainties / correlation plots for them at the request of reviewers. We did not include those results in the statistics, including Elo score computation, since it is unclear to us what type of statistical uncertainty does correctness on this dataset corresponds to. This is motivated by the fact, that all questions of TruthfulQA are intentionally designed in a fashion that either one or the other option will be dominant depending on specific characteristics of the model pretraining (i.e. data curation, order of sequence sampling, etc). This is in contrast with the relatively clear cut cases of other considered datasets, i.e. information retrieval from prompt (CoQA) or from weights (TriviaQA).

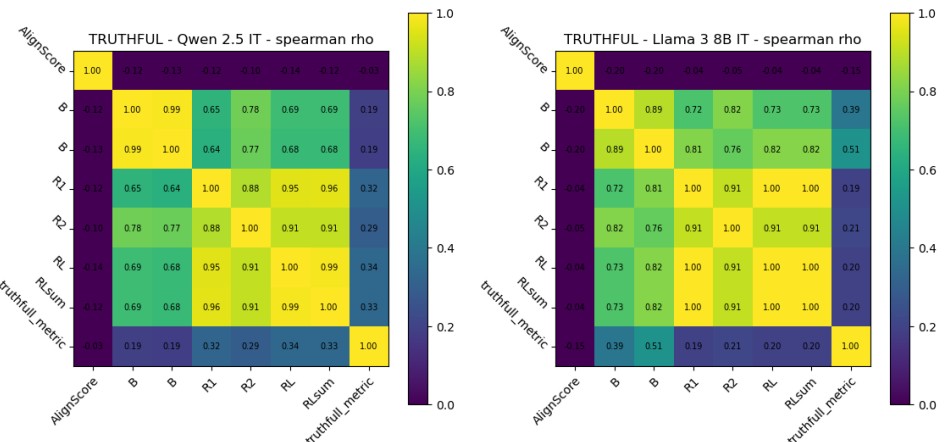

Figure 4: Agreement of ordering UE methods on TruthfulQA.

**Constrained Text Generation.** We used COLLIE (Yao et al., 2024) as a constrained text generation dataset. The challenges in this dataset are derived from passages of text from curated sources to ensure that the problems have solutions. Unlike many other constrained text datasets, COLLIE does not rely on judge evaluation.

**Code Completion.** Code completion problems allow for non-parametric correctness verification by means of unit tests. There are several popular public datasets for code completion (Austin et al., 2021; Chen et al., 2021; Hendrycks et al., 2021; Li et al., 2022) that feature an adequate test coverage rate. In our experiments we used BigCodeBench dataset in completion mode (Zhuo et al., 2024). The problems are all in python code, require no non-standard libraries and could be considered to be in distribution with respect to the training sets of the modern language models. To evaluate the exact solution, we used the remote endpoint provided by the creators of the dataset that automatically provides the per output and summary statistics. The labels were binarized based on fulfillment of all unit tests for each sample.

**OOD Datasets.** Unlike the image domain, obtaining OOD examples for text data is difficult. When thinking of OOD examples for text, one would imagine questions about things that have not yet come to be or are otherwise unknown or ambiguous in the general text corpora. Several datasets seek to provide artificial OOD examples. Known-Unknowns (Amayuelas et al., 2024) seek to collect questions that can be assumed to have controversial answers in the common training sets. SQuADv2 (Rajpurkar et al., 2018) provides questions formulated to be unanswerable given the prompt. Our search revealed only these two datasets as such that provide suitable OOD labels. We do not rule out the possibility that there are more such datasets in existence, but OOD detection is not a main focus of our work.

### B.4 DETAILS ON THE APPROXIMATE CORRECTNESS FUNCTIONS

**n-gram matching.** The standard n-gram matching correctness algorithms are the ROUGE (Lin, 2004) and BLEU families (Papineni et al., 2002). To turn these into correctness functions, one is

required to specify a threshold $d$ and the n-gram parameter $n$, so $\boldsymbol{\theta}_c = (d, n)$. In practice these algorithms often lack robustness and face criticism (Schluter, 2017).

**Embedding Based Correctness.** Learned correctness functions, such as BERTScore (Zhang et al., 2020) and BLEURT (Sellam et al., 2020) use similarity of the answer and the reference in an embedding space. Specifically, BERTScore computes contextual embeddings, calculates cosine similarity and applies F1 score. BLEURT uses the model to predict similarity directly. Both metrics could not be computed for longer generations due to context length limitations. AlignScore (Zha et al., 2023) uses a model fine-tuned to perform information alignment in order to assess similarity between the given and reference answers.

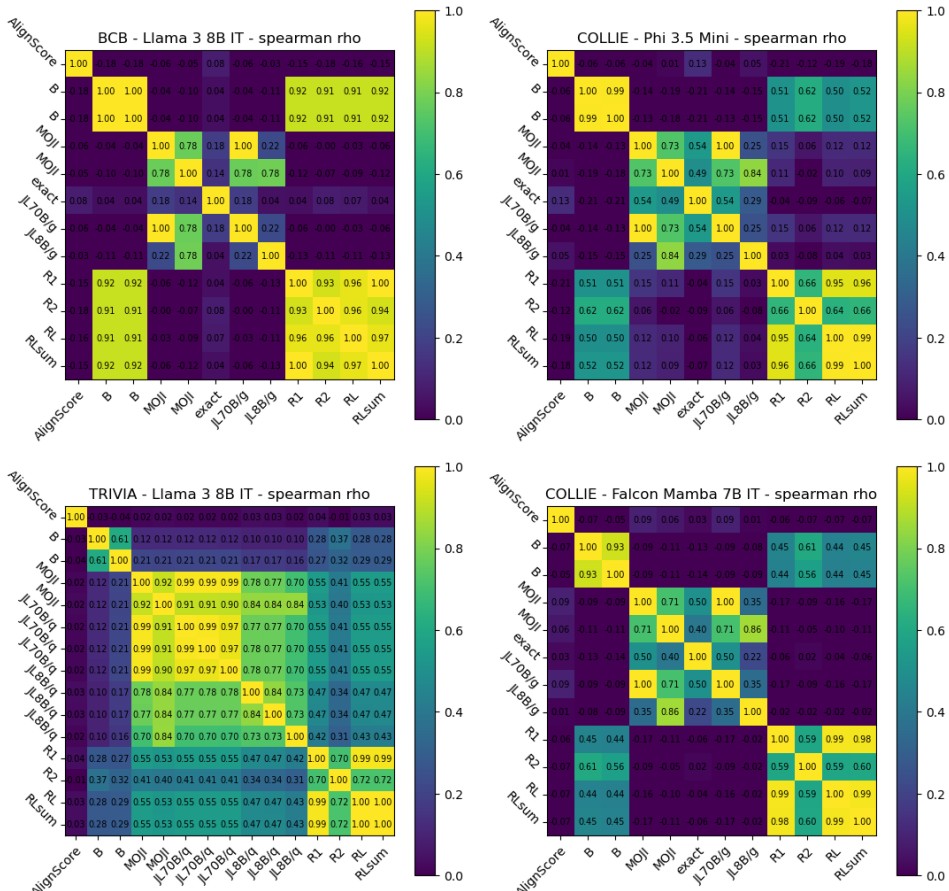

Figure 5: UE method ordering heatmaps including AlignScore. In our experiments it was always observed to be in its own cluster. Although the default settings from the AlignScore repository together with 'base' model were used, it is quite possible that some adjustments would improve its correlation to other correctness methods.

**LLM-as-a-judge.** LLM-as-a-judge (Zheng et al., 2023) prompts an LLM to confirm the correctness of the answer with respect to the reference. A model is provided with a prompt, question, proposed answer and reference answer. The prompt usually requests the model to generate yes if the question-answer-reference answer tuple is correct. While recently reasoning judges have been proposed, we utilized the usual judge models due to simplicity and noticeably lower evaluation costs.

### B.4.1 NOTES ON ROUGE-2 AND BLEU IMPLEMENTATION ARTIFACTS

Notably, ROUGE-2 and BLEU show low agreement to other n-gram based metrics while showing some higher than average agreement to each other. Low agreement of BLEU to other metrics can in

Table 3: Accuracies of the models for evaluated datasets according to corresponding correctness functions. This table lists the dataset / model papers evaluated in this work. Nan values in SQUAD is expected behavior, as there are no correctness labels for the artificially unanswerable OOD part. Known-Unknown (Amayuelas et al., 2024) dataset generations were performed without accuracy computation as we used it strictly as an OOD detection dataset.

| Dataset | Model / Temp | NaN Values | Accuracy | Main Correctness Function |
|---------|--------------|------------|----------|---------------------------|
| BCB | Llama-3 8b / 1. | 0 | 0.34 | Exact |
| BCB | Llama-3 8b IT / 1. | 2 | 0.21 | Exact |
| COLLIE | Phi-3.5 / 1. | 0 | 0.30 | Exact |
| COLLIE | Llama-3 70b IT / 1. | 0 | 0.49 | Exact |
| COLLIE | Falcon Mamba / 1. | 0 | 0.14 | Exact |
| COLLIE | Llama-3 8b / 1. | 0 | 0.145 | Exact |
| COLLIE | Falcon Mamba IT / 1. | 0 | 0.166 | Exact |
| COLLIE | Llama-3 8b IT / 1. | 0 | 0.42 | Exact |
| COLLIE | Phi-3.5 IT / 1. | 0 | 0.21 | Exact |
| COQA | Llama-3 8b IT / 1. | 0 | 0.86 | MoJI |
| COQA | Phi-3.5 IT / 1. | 0 | 0.81 | MoJI |
| COQA | Llama-3 8b / 1. | 0 | 0.54 | MoJI |
| COQA | Llama-3 70b / 1. | 0 | 0.73 | MoJI |
| SQUAD | Phi-3.5 IT / 1. | 5945 | 0.92 | MoJI |
| SQUAD | Llama-3 8b IT / 1. | 5945 | 0.94 | MoJI |
| SQUAD | Llama-3 8b / 1. | 5945 | 0.74 | MoJI |
| SQUAD | Llama-3 70b IT / 1. | 5945 | 0.94 | MoJI |
| TRIVIA | Phi-3.5 IT / 1. | 0 | 0.58 | MoJI |
| TRIVIA | Llama-3 8b IT / 1. | 0 | 0.74 | MoJI |
| BCB | Qwen2.5 32b IT / 0.6 | 2 | 0.174 | Exact |
| BCB | Llama-3 70b IT / 0.6 | 1 | 0.352 | Exact |
| BCB | Qwen2.5 7b IT / 0.6 | 3 | 0.069 | Exact |
| SQUAD | Qwen2.5 32b IT / 0.6 | 5945 | 0.959 | MoJI |
| COQA | Qwen2.5 32b IT / 0.6 | 0 | 0.835 | MoJI |
| COLLIE | Qwen2.5 32b IT / 0.6 | 0 | 0.436 | Exact |
| COLLIE | Qwen2.5 7b IT / 0.6 | 0 | 0.286 | Exact |

part be explained by correctness values being low, making the commonly used $0.5$ threshold a poor choice. Upon closer inspection it turned out that the standard implementations of ROUGE and BLEU that is widely used in uncertainty estimation evaluation (Luong et al., 2017) return correctness of zero if either the proposed or reference answers are shorter than a predefined n-gram, which is 2 for ROUGE-2 and 4 for BLEU. Considering the distribution of reference answers in QA datasets, this is a major artifact demanding attention.

## B.5 DETAILS ON EXPERIMENTAL SETTING

In our experiments we have preferred smaller models from diverse state of the art open source model families. Throughout our investigations, we use the Llama-3 8B and 70B (Dubey et al., 2024), Phi-3.5 (Abdin et al., 2024), Qwen2.5 (Qwen et al., 2025) and Falcon Mamba 7B (Zuo et al., 2024) series of models, both pretrained and instruction tuned (IT). Falcon Mamba models, although less performant than their attention based counterparts, were utilized to broaden the evaluation coverage to the upcoming linear attention models. The dataset model pairs considered and the accuracies achieved on the most appropriate correctness metric are listed in Tab. 3. The accuracies were evaluated on sequences generated with beam search ($n = 10$).

Computational resources required to perform generation and uncertainty computation are estimated to be in the range of 600 GPU-hours.

### B.5.1    STANDARD DEVIATION OF PERFORMANCE ESTIMATOR

The data for the Fig. 2 was generated in the following way. The judge calls were made preemptively, 14 per example for each selected dataset / model combination. The performance estimates $\xi_{SP-J}$ was computed for each judge. Then for each $n \in \{1 \ldots 14\}$ we uniformly sampled $\xi_{SP-J}$s $n$ times with replacement. The bootstrap SD was computed from 100 such samples. The SD was chosen over variance since it is simpler to interpret in terms of confidence interval and shares the AUROC performance unit.

### B.5.2    COMPUTING THE ELO RATING

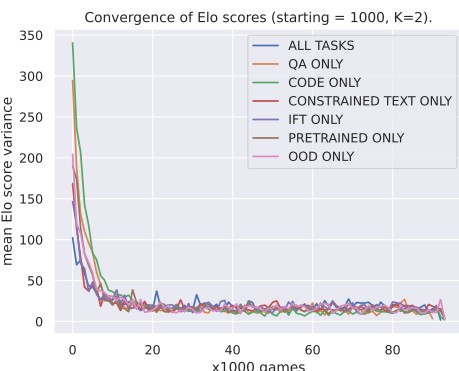

Figure 6: Convergence of Elo ratings on the various experimental subsets.

The Elo rating was computed as follows: the initial rating were initialized to 1000 for each method. For each step, a dataset / model pair was selected, as well as two distinct uncertainty estimation methods. Out of the two methods, one with higher AUC against the corresponding risk indicator would be considered the winner. The scores would then be updated according to the standard Elo update rule with $s = 400$ and $K = 2$. $K$ value roughly corresponds to the update step size modifier. The relatively low value of $K$ was selected since the optimization was performed for $100,000$ steps until convergence (Fig. 6). The mean and variance of the Elo scores over the last 1000 iterations were taken as the final values presented in Fig. 3.

### B.5.3    GRID DEFINITION FOR THE CORRECTNESS HACKING EXPERIMENT

For the correctness hacking table, the following parameter grid was made available: Rouge-L, Rouge-L-Sum, LLM-as-a-judge (Llama 70B QA prompt) with possible thresholds of 0.3 and 0.5 (relevant for Rouge family of methods). We have deliberately excluded methods such as BLEU, BLEURT, and ROUGE1/2 from this experiment, since they are not frequently used in the literature. Despite such restriction, the differences in Top-3 occurrences in Tab. 2 are substantial.

### B.5.4    ADDITIONAL SMALL SCALE HUMAN ANNOTATION

In order to assess the general ability of MoJI to provide good quality labels, we have performed selective annotations of 300 question-answer pairs from the CoQA - Llama3 8B IT model rollouts. 3 humans annotated the examples independently upon agreeing to the basic rules (i.e. that if the reference answer contradicts the text, then it is not to be considered and instead the answer the model answer is to be assessed with respect to the provided text). Overall the human judges reached perfect agreement on all but 31 (10%) questions out of 300. 9 questions were deemed to contain an inadequate reference answer.

The correlation of MoJI and the Judges to human generated labels is shown in Fig. 7. It can be noted, that MoJI correlation to human annotators is often on par with correlation between the annotators. It is also higher than the average correlation of judges to humans. This indicate that MoJI generally improves the quality of correctness labels compared to individual judges.

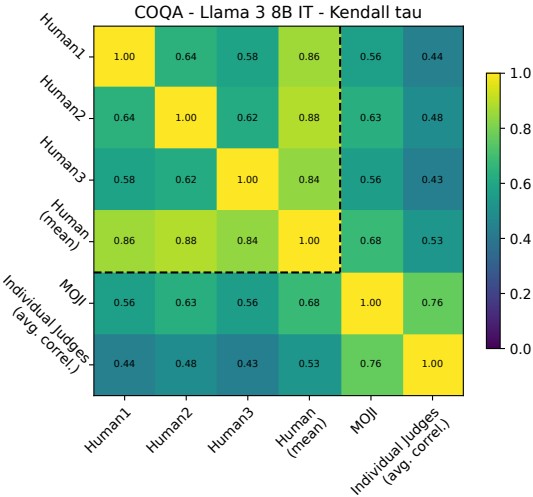

Figure 7: Correlation of MoJI to human annotators on a sample of 300 question-answers from CoQA. MoJI shows higher correlation to human annotators than individual judges on average.

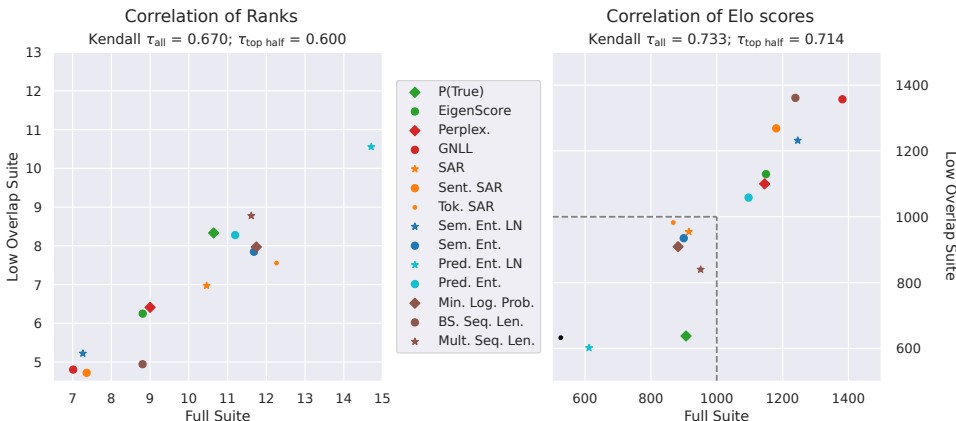

Figure 8: Average Rank (Left) vs Elo score (Rank) evaluated for the complete and synthetic indirect comparison scenarios. For ranks: lower is better, for Elo scores higher is better.

### B.5.5 ABLATIONS OF ELO SCORE

**Indirect Comparison** To demonstrate the advantage of Elo Score for indirect comparison we have occluded parts of the experimental suite in the following fashion:

1. For all Llama models only the following UE methods were available: Pred.Ent., Pred.Ent. LN, Seq.Len., BS Seq.Len., Token SAR, Sent. SAR, SAR, Perplex., Sem. Ent., Min. Log. Prob.

2. For the remaining models only the following UE methods are available: Sent. SAR, SAR, Perplex., Sem. Ent., Min. Log. Prob., Sem.Ent. LN, P(True), GNLL, EigenScore.

The methods were chosen arbitrarily (using python list slicing [:10] and [5:] on the list of names). A relatively small 'bridge' of methods was left as the partial overlap. We then aggregate the 'low overlap' and full experimental suite using the average rank (similar to Vashurin et al. (2025)) and the Elo score. The results are provided in Fig. 8. We demonstrate that Elo score is more robust at retaining the ordering of methods when subjected to such 'low overlap' conditions, in particular on the higher end of performance spectrum.

This overlap condition is relatively mild, as more invasive options could be devised (e.g. breaking down across the datasets).

## C    APPLICATION DETAILS FOR JUDGE MODELS

We used judge models were of Llama-3 family with 8B (earlier experiments only), 70B and 405B parameters. In order to achieve marginalization over the model class in MoJI, we further added Qwen2.5 and Deepseek-v3 models.

**Sampling Parameters**    The length of the completion was observed to be largely in 2-3 token range, indicating that the prompting largely succeeded at imposing anticipated output structure onto the model. The models were evaluated at temperatures of 0.5 and 1..

**Judge Models Used**    Tab. 4 lists the judges, prompt temperature combinations used in our experiments. We used two different prompts mostly to showcase the difference between the outcomes on the COLLIE dataset. If yes was among the words generated in the answer, the correctness of 1 was assigned. This is consistent with usage in (Farquhar et al., 2024; Aichberger et al., 2024). Imposing a more rigid structure upon the answers (i.e. guided generation (Dong et al., 2024)) is left to future work. The use of reasoning judges is relatively novel and has a major drawback of being substantially more expensive and further complicating the answer extraction. This would be especially strongly felt in correctness evaluation for relatively short QA answers. Therefore we deem it to be beyond the scope of our work.

Table 4:  Judge model configurations used for correctness prediction.  QA and Gen prompts are provided in Apx.Sec.C.2. Additionally, multiple samples were taken from each judge model.

| Model | Prompt | Temperature |
| --- | --- | --- |
| Llama3 405B | gen | 1.0 |
| Llama3 405B | qa | 0.5 |
| Llama3 405B | qa | 1.0 |
| Llama3 70B | gen | 0.5 |
| Llama3 70B | gen | 1.0 |
| Llama3 70B | qa | 0.5 |
| Llama3 70B | qa | 1.0 |
| Llama3 8B | gen | 0.5 |
| Llama3 8B | gen | 1.0 |
| Llama3 8B | qa | 0.5 |
| Llama3 8B | qa | 1.0 |
| Qwen 32B | gen | 0.5 |
| Qwen 32B | qa | 0.5 |
| Qwen 7B | gen | 0.5 |
| Qwen 7B | qa | 0.5 |
| Deepseek v3 | gen | 0.5 |

### C.1    COMPUTATIONAL BURDEN OF MULTIPLE JUDGES

The judge models were operated via an API in 8 bit floating point precision without a fixed random seed. A total computational burden of evaluating the judge models for our experimental suite is estimated to be under 40M tokens, predominantly in prefill mode due to really short answer length required. This translated into $\approx 40\$$ of API calls.

Judge model evaluation implies short generation length, on the order of several tokens, as has been observed by us. This makes it fast relative to the original sequence generation, since the prefill is $10 - 100$ times faster per token than generation. This is a major advantage of classic LLM-as-a-judge approaches compared to their reasoning counterparts.

We have evaluated a variable number of judges per experiment setting (see Fig. 11). Based on Fig. 2 having a MoJI ensemble of 4 judges already bears great benefit, while more than 10 judge calls might be redundant.

### C.2 PROMPTS USED FOR JUDGE MODELS

QA prompt follows the implementation of Farquhar et al. (2024):

```
We are assessing the quality of answers
to the following question: {question}
The expected answer is: {correct_answer}.
The proposed answer is: {predicted_answer}
Within the context of the question,
does the proposed answer mean the same as the expected answer?
Respond only with yes or no.
Response:
```

Gen prompt is derived from the QA prompt with minor modifications:

```
We are assessing the quality of answers
to the following question: {question}
The following are example answers: {correct_answer}.
The proposed answer is: {predicted_answer}
Within the context of the question and example answer,
is the proposed answer correct?
Respond only with yes or no.
Response:
```

## D   ADDITIONAL THEORETICAL CONSIDERATIONS

### D.1   EMPIRICAL PROPERTIES OF UNCERTAINTY ESTIMATION ALGORITHMS

According to the how uncertainty quantification algorithms are evaluated in the literature (Welling & Teh, 2011; Gal & Ghahramani, 2016; Lakshminarayanan et al., 2017; Malinin & Gales, 2018; D' Angelo & Fortuin, 2021; Daxberger et al., 2021; Mukhoti et al., 2023; Schweighofer et al., 2023), we can say that uncertainty is a function $\hat{u}_{\text{ale}}(\boldsymbol{x}, \boldsymbol{w}; \boldsymbol{\theta}_u)$ (aleatoric) or $\hat{u}_{\text{epi}}(\boldsymbol{x}, \mathcal{D}; \boldsymbol{\theta}_u)$ (epistemic) with positive real valued codomain that has the following empirical properties:

1. $\hat{u}$ is higher for $\boldsymbol{x}' \sim \mathcal{D}_{\text{test}}$ than for $\boldsymbol{x} \sim \mathcal{D}_{\text{test}}$ if the risk of prediction using $\boldsymbol{w}$ (aleatoric) or $\boldsymbol{w} \sim p(\boldsymbol{w} \mid \mathcal{D})$ (epistemic) for $\boldsymbol{x}'$ is higher than for $\boldsymbol{x}$.

2. $\hat{u}$ is not lower for $\boldsymbol{x}'$ than for $\boldsymbol{x} \sim \mathcal{D}_{\text{test}}$ if $\boldsymbol{x}'$ is drawn from a different data generating function than one that produced the training data $\mathcal{D}$.

3. $\hat{u}$ is not lower for $\boldsymbol{x}'$ than for $\boldsymbol{x} \sim \mathcal{D}_{\text{test}}$ if $\boldsymbol{x}'$ is obtained from $\boldsymbol{x}$ by some perturbation.

These properties can be distilled from ubiquitously used evaluation protocols in uncertainty quantification literature in classification setting. Note that the first and third properties are characteristic of both aleatoric and epistemic uncertainty, whereas the second is usually attributed to epistemic uncertainty. In the classification setting, most of the literature is focused on epistemic uncertainty since it involves, depending on the definition, estimating a more difficult posterior integral of a divergence, which requires intricate posterior sampling techniques (Wilson & Izmailov, 2020; Schweighofer et al., 2023). The three empirical properties can be unified in terms of viewing uncertainty as an indicator of prediction risk (Kotelevskii & Panov, 2025; Lahlou et al., 2023).

Another assumption is sometimes used:

4. If $\hat{u}_{epi}$ is higher for $\boldsymbol{x}' \sim \mathcal{D}_{\text{domain}}$ than for $\boldsymbol{x}$, then adding the $(\boldsymbol{x}', \boldsymbol{y}')$ to the training dataset $\mathcal{D}$ would on expectation lead to higher risk reduction on $\mathcal{D}_{\text{domain}}$ than adding $(\boldsymbol{x}, \boldsymbol{y})$.

This is the active learning assumption which in classification literature is usually associated with the epistemic uncertainty (Kirsch, 2024). Active Learning evaluation is a challenging task with many caveats even in the classification setting (Hacohen et al., 2022; Lüth et al., 2023). Furthermore it requires a true label and some degree of model tuning. Autoregressive generation further complicates this mode of evaluation. Therefore we do not consider AL assumption for evaluation in our work.

The way these assumptions are formulated implies that the correlation coefficient according to which they are evaluated must be invariant to monotone increasing transformations.

## D.2 EFFECTS OF REFERENCE LABEL PERTURBATION ON RANK CORRELATION

In this section we investigate the effects of the defects of the reference class labels on rank correlation. We specifically focus on AUC, as it is the rank correlation most commonly used in Uncertainty Estimation literature for risk correlation experiments. We show that both variance and bias in risk indicator values lead to biased AUC estimates. Both of the considered scenarios support using MoJI as the approximate correctness measure of choice.

### D.2.1 SAMPLE AUROC

Sample AUROC can be computed explicitly as follows:

$$\text{AUC}^{\text{s}} = \frac{1}{n_0 n_1} \sum_{i:y_i=1} \sum_{j:y_j=0} \mathbb{1}(s_i > s_j) + 0.5 \cdot \mathbb{1}(s_i = s_j) \tag{18}$$

It has an equivalent MC estimator that implies sampling positive-negative labeled pairs:

$$\text{AUC}^{\text{s-MC}} \approx \frac{1}{M} \sum_{i}^{M} \mathbb{1}(s_i^1 > s_i^0) \tag{19}$$

The two forms are equivalent and are unbiased and consistent AUC estimators and are equivalent to the original rank based U statistic (Mann & Whitney, 1947). Generally, the AUC corresponds to the expected probability that the scorer $s : \mathcal{X} \to \mathbb{R}$ ranks the items $(x_1, \dots x_n)$ in a way that those with positive binary labels $(y_1, \dots y_n)$ have higher score than ones with negative labels.

$$\text{AUC} = \text{E}_{x_p \sim p(x,y=0)} \text{E}_{x_n \sim p(x,y=1)} P\left[s(x_p) > s(x_n)\right] \tag{20}$$

In case of empirical assessment of the uncertainty estimation algorithm by correlation to risk (as per Sec. 2 and Apx. D.1) $\xi$, the $y$ labels are the negated correctness $\neg c$ and scores are the uncertainty estimates $\hat{u}$.

**Sample AUROC with label noise**  Let us now consider scenario, where the reference labels are perturbed randomly by a Bernoulli noise:

$$c_{x_i}^{\text{noisy}} = \begin{cases} c_{x_i} & \text{if } \gamma \sim \mathcal{B}(p) = 0 \\ \neg c_{x_i} & \text{if } \gamma \sim \mathcal{B}(p) = 1 \end{cases} \tag{21}$$

Note, that rounded expectation of $c_{x_i}^{\text{noisy}}$ (its median) equals the true value of $c_{x_i}$ if the noise magnitude $p < 0.5$:

$$\text{round}\left[\text{E}[c^{\text{noisy}}(x_i)]\right] = c(x_i) \tag{22}$$

Informally this can be viewed as an unbiased estimator of $c(x_i)$ with added variance for a binary variable. $\gamma$ is independent of the example $i$ to which it applies, contrary to the bias introduced by distortion in the previous section.

To inspect the properties of the AUC estimate in case of of noisy reference, we will use the $\text{AUC}^{\text{MC}}$ from Eq. (19) formulation of the estimator, as the direct sample AUC estimation from Eq. (18) is less

suitable for accommodating the noise term. In this regime we require sampling pairs of inputs with positive/negative label $i$. This assumes ability to specifically sample the positive or negative class, which we take for granted (i.e. class balance assumption) without additional importance sampling considerations. We decompose the Eq. (19) similarly to what we did for the bias case:

$$\text{AUC}^{\text{noisy-MC}} =$$

$$= \frac{1}{M} \sum_i^M \mathbb{1}\left(s(x_i^a) > s_{(}x_i^b)\mid c^{\text{noisy}}(x_i^a) = 1, c^{\text{noisy}}(x_i^b) = 0\right)$$

$$= \frac{1}{M} \sum_i^M \begin{cases} \mathbb{1}\left(s(x_i^a) > s_{(}x_i^b)\mid c(x_i^a) = 1, c(x_i^b) = 0\right) \cdot p(\gamma = 1)^2 + \\ \mathbb{1}\left(s(x_i^a) > s_{(}x_i^b)\mid c(x_i^a) = 1, c(x_i^b) = 1\right) \cdot p(\gamma = 0)p(\gamma = 1) + \\ \mathbb{1}\left(s(x_i^a) > s_{(}x_i^b)\mid c(x_i^a) = 0, c(x_i^b) = 0\right) \cdot p(\gamma = 0)p(\gamma = 1) + \\ \mathbb{1}\left(s(x_i^a) > s_{(}x_i^b)\mid c(x_i^a) = 0, c(x_i^b) = 1\right) \cdot p(\gamma = 0)^2 \end{cases}$$

$$= \frac{1}{M} \sum_i^M \begin{cases} \mathbb{1}\left(s(x_i^a) > s_{(}x_i^b)\mid c(x_i^a) = 1, c(x_i^b) = 0\right) \cdot (1-p)^2 + \\ \mathbb{1}\left(s(x_i^a) > s_{(}x_i^b)\mid c(x_i^a) = 1, c(x_i^b) = 1\right) \cdot p \cdot (1-p) + \\ \mathbb{1}\left(s(x_i^a) > s_{(}x_i^b)\mid c(x_i^a) = 0, c(x_i^b) = 0\right) \cdot p \cdot (1-p) + \\ \mathbb{1}\left(s(x_i^a) > s_{(}x_i^b)\mid c(x_i^a) = 0, c(x_i^b) = 1\right) \cdot p^2 \end{cases} \tag{23}$$

Note that the coefficients in 23 sum up to 1, which makes sense. Then we can proceed by separating the part that corresponds to the AUC estimator with unbiased labels:

$$\text{AUC}^{\text{noisy-MC}} =$$

$$= \frac{1}{M} \sum_i^M \mathbb{1}\left(s(x_i^a) > s(x_i^b) \mid c^{\text{noisy}}(x_i^a) = 1, c^{\text{noisy}}(x_i^b) = 0\right)$$

$$= \frac{1}{M} \sum_i^M \mathbb{1}\left(s(x_i^a) > s(x_i^b) \mid c(x_i^a) = 1, c(x_i^b) = 0\right) \cdot (1-p)^2 +$$

$$+ \frac{1}{M} \sum_i^M \begin{cases} \mathbb{1}\left(s(x_i^a) > s(x_i^b) \mid c(x_i^a) = 1, c(x_i^b) = 1\right) \cdot p \cdot (1-p) + \\ \mathbb{1}\left(s(x_i^a) > s(x_i^b) \mid c(x_i^a) = 0, c(x_i^b) = 0\right) \cdot p \cdot (1-p) + \\ \mathbb{1}\left(s(x_i^a) > s(x_i^b) \mid c(x_i^a) = 0, c(x_i^b) = 1\right) \cdot p^2 \end{cases}$$

$$= \text{AUC}^{\text{MC}} \cdot (1-p)^2 +$$

$$+ \frac{1}{M} \sum_i^M \begin{cases} \mathbb{1}\left(s(x_i^a) > s(x_i^b) \mid c(x_i^a) = 1, c(x_i^b) = 1\right) \cdot p \cdot (1-p) + \\ \mathbb{1}\left(s(x_i^a) > s(x_i^b) \mid c(x_i^a) = 0, c(x_i^b) = 0\right) \cdot p \cdot (1-p) + \\ \mathbb{1}\left(s(x_i^a) > s(x_i^b) \mid c(x_i^a) = 0, c(x_i^b) = 1\right) \cdot p^2 \end{cases}$$

$$= \text{AUC}^{\text{MC}} \cdot (1-p)^2 +$$

$$+ \frac{1}{M} \sum_i^M \begin{cases} \mathbb{1}\left(s(x_i^a) > s(x_i^b) \mid c(x_i^a) = 1, c(x_i^b) = 1\right) \cdot p \cdot (1-p) + \\ \mathbb{1}\left(s(x_i^a) > s(x_i^b) \mid c(x_i^a) = 0, c(x_i^b) = 0\right) \cdot p \cdot (1-p) + \\ \left(1 - \mathbb{1}\left(s(x_i^a) > s(x_i^b) \mid c(x_i^a) = 1, c(x_i^b) = 0\right)\right) \cdot p^2 \end{cases}$$

$$= \text{AUC}^{\text{MC}} \cdot (1-p)^2 + \frac{1}{M} \sum_i^M p^2 - \text{AUC}^{\text{MC}} \cdot p^2 +$$

$$+ \frac{1}{M} \sum_i^M \begin{cases} \mathbb{1}\left(s(x_i^a) > s(x_i^b) \mid c(x_i^a) = 1, c(x_i^b) = 1\right) \cdot p \cdot (1-p) + \\ \mathbb{1}\left(s(x_i^a) > s(x_i^b) \mid c(x_i^a) = 0, c(x_i^b) = 0\right) \cdot p \cdot (1-p) \end{cases}$$

$$= \text{AUC}^{\text{MC}} \cdot (1-2p) + \frac{1}{M} \sum_i^M p^2 +$$

$$+ p \cdot (1-p) \frac{1}{M} \sum_i^M \begin{cases} \mathbb{1}\left(s(x_i^a) > s(x_i^b) \mid c(x_i^a) = 1, c(x_i^b) = 1\right) + \\ \mathbb{1}\left(s(x_i^a) > s(x_i^b) \mid c(x_i^a) = 0, c(x_i^b) = 0\right) \end{cases}$$

$$\tag{24}$$

We can safely assume that the two identity terms within the classes sum up to $0.5$ over large number of samples. This is because within the same class we can sample both $(x_i^a, x_i^b)$ and $(x_i^b, x_i^a)$ with the same likelihood.

$$\text{AUC}^{\text{noisy-MC}} =$$

$$= \text{AUC}^{\text{MC}} \cdot (1-2p) + \frac{1}{M} \sum_i^M p^2 + \sum_i^M \frac{1}{M} p \cdot (1-p)$$

$$= \text{AUC}^{\text{MC}} \cdot (1-2p) + p \tag{25}$$

While in Eq. (25) the first term is lower than the value obtained with unbiased labels by a factor of $1 - 2p$. With this, $\text{AUC}^{\text{noisy-MC}} = \text{AUC}^{\text{MC}}$ only when the $\text{AUC}^{\text{MC}} = 0.5$. Intuitively, we can see that random classifier will not be affected by noise in the labels. This shows, that ultimately, introducing random noise to the labels increases the bias of the AUC estimator and results in a loss of its asymptotic consistency. *In context of our work* this demonstrates that having variance in the risk indicator (i.e. stochastic approximate correctness) yields a biased estimate of $\xi$ when using AUC as rank correlation. This is particularly relevant to samples from LLM-as-a-judge, which are rolled out stochastically.

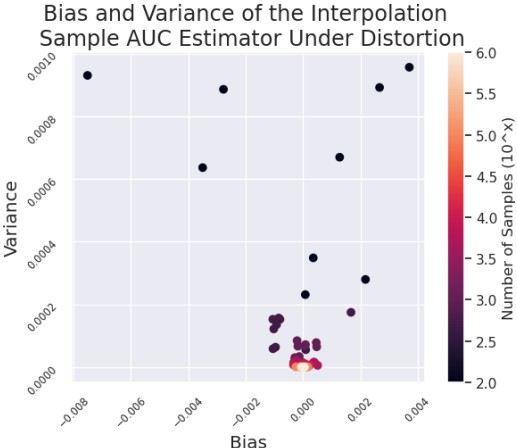

Figure 9: Empirical verification of Eq. (27). This was conducted with synthetic random data with initial labels initialized for AUROC scores of 0.6, 0.75, 0.9 with distortion rates of 0.1, 0.2 and 0.3. 120 evaluations were made for each point to collect the corresponding statistics. The decomposition has no practical bias and low variance, particularly for sample sizes of $10^3$ which corresponds to common sizes of QA datasets.

**Sample AUROC with biased labels**  Lets consider a scenario, where the correctness function is biased. This is equivalent to permanently perturbing the correctness labels $(c_1, \ldots c_n)$ with some distortion function $d : X \mapsto \{0, 1\}$:

$$c_{x_i}^{\mathrm{b}} = \begin{cases} c_{x_i} & \text{if } d_{x_i} = 0 \\ \neg c_{x_i} & \text{if } d_{x_i} = 1 \end{cases} \tag{26}$$

For brevity, we refer to $c_{x_i}^{\mathrm{b}}$ as $c_i$ and to $d_{x_i}$ as $d_i$. Then (ignoring the ties for simplicity):

$$\mathrm{AUC}^{\text{s-b}} =$$

$$= \frac{1}{n_0 n_1} \sum_{i:y_i=1} \sum_{j:y_j=0} \mathbb{1}(s_i > s_j)$$

$$= \frac{1}{n_0} \sum_{i:y_i=1} \begin{cases} p(d_j = 0, y_j = 0) \sum_{j:y_j=0 \wedge d_j=0} \mathbb{1}(s_i > s_j) + \\ p(d_j = 1, y_j = 1) \sum_{j:y_j=1 \wedge d_j=1} \mathbb{1}(s_i > s_j) \end{cases}$$

$$= \begin{cases} \sum_{i:y_i=1 \wedge d_i=0} \sum_{j:y_j=0 \wedge d_j=0} \mathbb{1}(s_i > s_j) p(y_i = 1, d_i = 0) p(d_j = 0, y_j = 0) + \\ \sum_{i:y_i=0 \wedge d_i=1} \sum_{j:y_j=0 \wedge d_j=0} \mathbb{1}(s_i > s_j) p(y_i = 0, d_i = 1) p(d_j = 0, y_j = 0) + \\ \sum_{i:y_i=1 \wedge d_i=0} \sum_{j:y_j=1 \wedge d_j=1} \mathbb{1}(s_i > s_j) p(y_i = 1, d_i = 0) p(d_j = 1, y_j = 1) + \\ \sum_{i:y_i=0 \wedge d_i=1} \sum_{j:y_j=1 \wedge d_j=1} \mathbb{1}(s_i > s_j) p(y_i = 0, d_i = 1) p(d_j = 1, y_j = 1) \end{cases}$$

$$\approx \begin{cases} AUC^{\mathrm{s}} \cdot p(d_i = 0, d_j = 0) + \\ 0.5 \cdot (p(d_i = 1, d_j = 0) + p(d_i = 0, d_j = 1)) + \\ (1 - AUC^{\mathrm{s*}}) \cdot p(d_i = 1, d_j = 1) \end{cases}$$

$$= \begin{cases} AUC^{\mathrm{s}} \cdot \frac{n(d_i=0)n(d_j=0)}{n_0 n_1} + \\ 0.5 \cdot \left( \frac{n(d_i=1)n(d_j=0)}{n_0 n_1} + \frac{n(d_i=0)n(d_j=1)}{n_0 n_1} \right) + \\ (1 - AUC^{\mathrm{s}\,*}) \cdot \frac{n(d_i=1)n(d_j=1)}{n_0 n_1} \end{cases} \tag{27}$$

To obtain the Eq. (27) we first decompose the AUC estimate with distorted labels into 4 terms. These terms correspond to the possible cases of label perturbation combinations.

The first and the last term then can be expressed through the sample AUC with unbiased labels, which holds in the asymptotic case of large sample size ($N \to \inf$ where $N = n_0 + n_1$). The middle two terms equal $0.5$ by symmetry argument.

$\text{AUC}^{\text{s-b}} =$

$$= AUC^{\text{s}} \cdot \frac{n(d_i = 0)n(d_j = 0)}{n_0 n_1} + 0.5 \cdot \left( \frac{n(d_i = 1)n(d_j = 0)}{n_0 n_1} + \frac{n(d_i = 0)n(d_j = 1)}{n_0 n_1} \right) +$$

$$+ (1 - AUC^{\text{s}\,*}) \cdot \frac{n(d_i = 1)n(d_j = 1)}{n_0 n_1}$$

$$= AUC^{\text{s}} \cdot \frac{n(d_i = 0)n(d_j = 0)}{n_0 n_1} - AUC^{\text{s}\,*} \cdot \frac{n(d_i = 1)n(d_j = 1)}{n_0 n_1} +$$

$$+ 0.5 \cdot \left( \frac{n(d_i = 1)n(d_j = 0)}{n_0 n_1} + \frac{n(d_i = 0)n(d_j = 1)}{n_0 n_1} \right) + \frac{n(d_i = 1)n(d_j = 1)}{n_0 n_1}$$

$$= AUC^{\text{s}} \cdot \frac{n(d_i = 0)n(d_j = 0)}{n_0 n_1} - AUC^{\text{s}\,*} \cdot \frac{n(d_i = 1)n(d_j = 1)}{n_0 n_1} + \tag{28}$$

$$+ 0.5 \left( \frac{n(d_i = 1)}{n_0} + \frac{n(d_j = 1)}{n_1} \right) \tag{29}$$

Here the $AUC^{\text{s}\,*}$ is the AUC of the subsample with flipped labels and $AUC^{\text{s}}$ is the AUC of the undistorted part. Note, that in case of large sample size and random flipping of labels, this expression becomes equivalent to Eq. (25).

This shows, that the deviation from the original AUC depends on a) magnitude of distortion; b) on whether the AUC of distorted partition is similar to that of the undistorted partition. If the distortion is produced by random noise like in the previous section, the bias is higher if no resampling is done. This part of the identity above results in *In context of our work*, this shows that biased the risk indicator labels leads to bias and loss of consistency of the $\xi$ estimate compared to the case of unbiased indicator.

### D.2.2 Rank Correlation other than AUROC

Other popular rank correlation metric that is used is Spearman $\rho$. We leave derivation of the identities under conditions above for Spearman $\rho$ to future work. At the same time, since under discretized labels Spearman $\rho$ is numerically equivalent to AUROC, we would expect the identities for AUROC to look similar. Dorner et al. (2025) provides a discussion on binary vs non-binary labels for general purpose LM evaluation.

## E   Comparison of LLM-as-a-judge to Each other and Exact Correctness

In Fig. 11 we investigate the consistency of correctness assessment between different judge models. We can observe, that even identical models can diverge based on the prompt. When the temperature is not set to 0, we are additionally facing variability due to sampling outputs from the judge model.

### E.1   Selected Examples from QA Datasets where Judges Disagree

We have manually inspected some examples from TriviaQA, a commonly used QA dataset. We chose to manually inspect this dataset due to relative simplicity to establish causality due to short question and answer pairs which can be inspected quickly. We have selected top 15 examples with the highest MoJI entropy and inspected them manually. Most of the examples in the list contained obvious labeling problems with conflicting aliases. We present the examples with the most obvious ambiguity issues (from human standpoint) from the top 15 list, reporting their dataset id as well as their rank by MoJI entropy. The answers given by the model for which MoJI correctness was initially computed (Llama 3 8B IT in this case) are not provided as they are not relevant to the label noise inherent to the dataset.

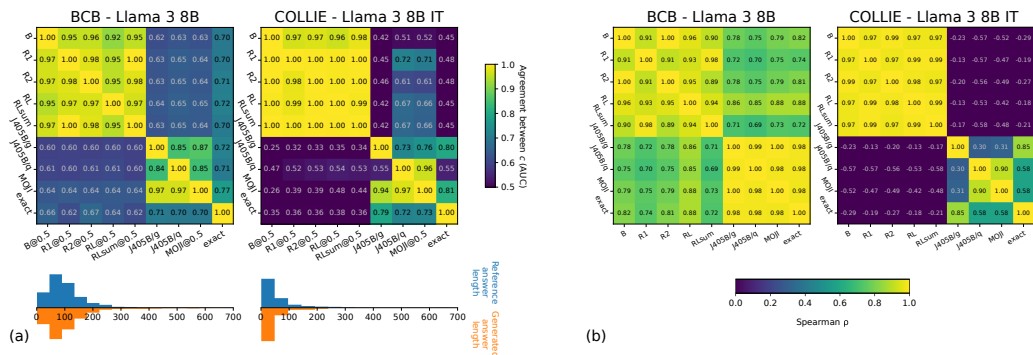

Figure 10: Correctness consistency on structured datasets. R indicates ROUGE family, B - BLEU. judge models are indicated with J, 'q' stands for QA prompt used in Farquhar et al. (2024) while 'g' stands for a more general prompt to evaluate correctness (see Apx. C.2 for more details on prompting). **(a)** Agreement of correctness metrics in terms of mutual AUROC (not symmetric). Column values are binarized at 0.5 where applicable. **(b)** Correlation of UE algorithm orderings when compared between corresponding pairs of correctness functions.

```
TRIVIAQA: 17181
MoJI Entropy rank: 2
question: Which Canadian born actor played an Irishman in
The Eagle Has Landed?,
 question_id: bt_1731,
 question_source: http://billturnbull.quiz4free.com/,
 answer: {aliases: [Sutherland (district),
   North West Sutherland,
   Cataibh,
   County of Sutherland,
   Sutherlandshire,
   Sutherland (local government district, Highland region),
   Sutherland,
   South East Sutherland],
  normalized_value: sutherland,
   value: Sutherland}
>> Confusing aliases, mixing the right labels with the wrong ones.

TRIVIAQA: 16858
MoJI Entropy rank: 8
question: Who became Prime Minister of Canada in November last year?,
 question_id: odql_12266,
 question_source: http://www.odquiz.org.uk/,
 answer: {aliases: [Trudeau, justin, Justin trudeau, Justin Trudeau],
  value: Justin Trudeau}
>> Temporal question the answer to which depends on time horizon
>> which was not specified.

TRIVIAQA: 7275
MoJI Entropy rank: 9
question: Kodkod, margay, oncilla and caracal are all types of what?,
 question_id: sfq_24575,
 question_source: www.sfquiz.org.uk,
 answer: {aliases: [(Wild) at],
  normalized_value: wild at,
   value: (Wild) at}
>> Obviously misextracted answer,
>> perhaps the correct labels was supposed to be 'Wild Cat'.
```

```
TRIVIAQA: 2881
MoJI Entropy rank: 11
 question: A Tale of Two Cities?,
 question_id: bb_2192,
 question_source: http://www.businessballs.com/,
 answer: {aliases: [Charles Dickons,
   C Dickens, Charles John Huffam Dickens, Dickens, Charles,
   Dickensian, Dickensian character, CJH Dickens,
   Charles Dickins, Charles John Huffam Dickens FRSA,
   Charles dickens, Dickens, Charels Dickens,
   Charles John Huffam Dickens, FRSA,
   Dickens charles, Charles Dickens],
  normalized_value: charles dickens,
  type: WikipediaEntity,
  value: Charles Dickens}
>> Vague, poorly posed question, unclear what is required.

TRIVIA: 15672
MoJI Entropy rank: 15
question: Which Scottish football team plays home games at Easter Road?,
 question_id: sfq_23124,
 question_source: www.sfquiz.org.uk,
 answer: {aliases: [The Hibernian,
   Charles Byrne (Journalist),
   HIBERNIAN],
  normalized_value: hibernian,
  value: HIBERNIAN},
>> Correct aliases mixed with irrelevant.
```

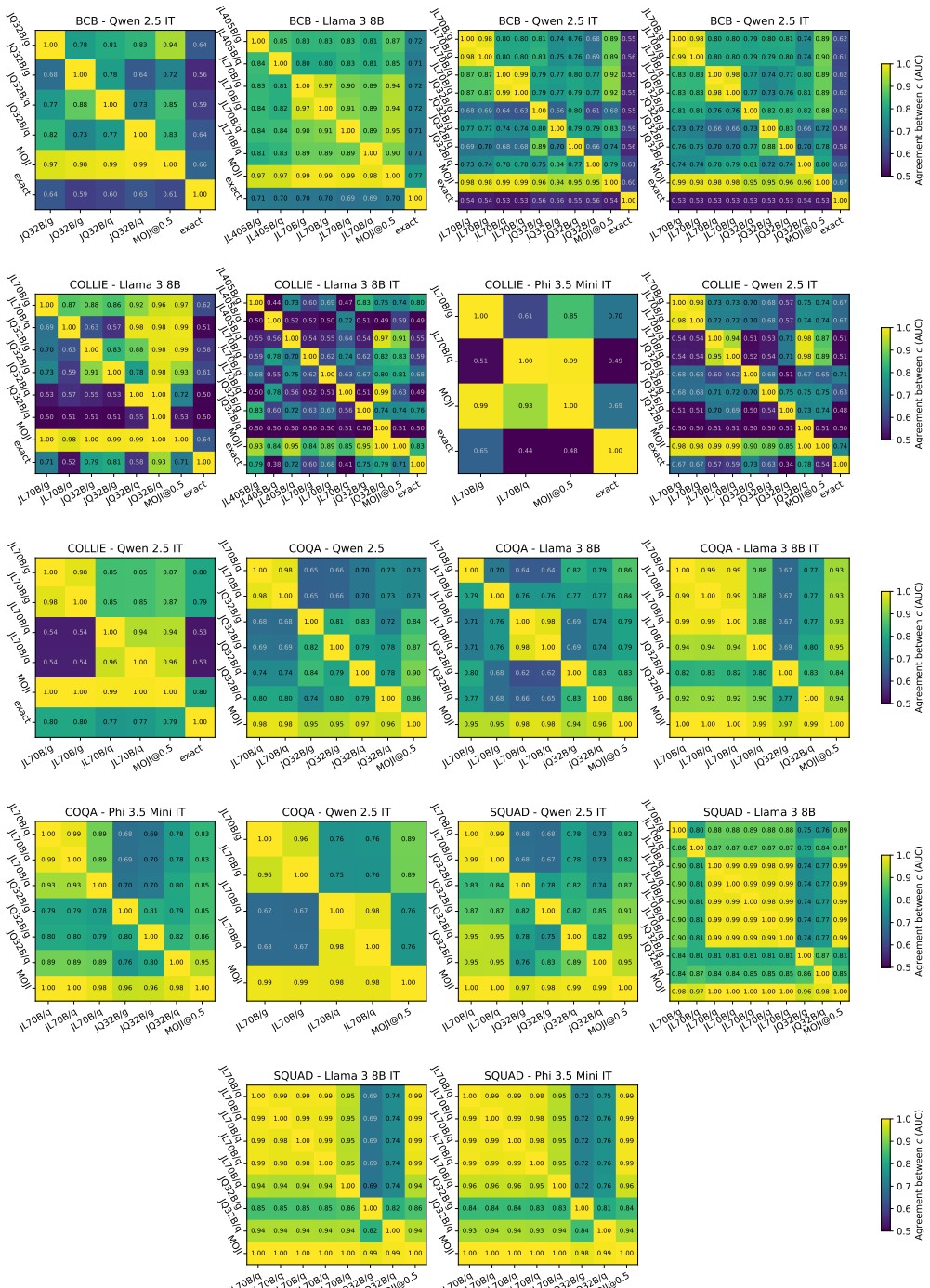

Figure 11: Agreement scores between judges on selected combinations. The ticks indicate model size / prompt / sampling temperature used to assess correctness. Judges of similar sizes tend to agree. Larger judge models tend to agree better with the exact solution, especially on COLLIE. The sampling temperature of the judge model appears to have a relatively minor effect on the outcome. Prompt affects the evaluation quality substantially, especially on COLLIE, which requires much less direct pattern matching and more implicit computation. Some judges appear more than once indicating multiple samples taken from the same judge configuration.

