# OpenReview forum: "Addressing Pitfalls in the Evaluation of Uncertainty Estimation Methods for Natural Language Generation"
_ICLR.cc/2026/Conference — ICLR 2026 Poster_

### Official Review · Reviewer_4ZXg · 2025-10-30

**Soundness:** 2
**Presentation:** 3
**Contribution:** 2
**Rating:** 4
**Confidence:** 3

**Summary:**

The paper shows that suboptimal NLG eval affects the evaluation of uncertainty quantifiers designed for parameterising decision making pipelines (such as selective prediction and OOD detection) using LLMs.

The paper shows that different automated proxies for NLG eval lead to different patterns of correlation between UQ and risk in decision making, with high disagreement across the available options. The paper shows that stochasticity inherent to an NLG evaluator (due to predictive uncertainty and/or effects of different prompts) too play a role.

The paper proposes to aggregate statistics from not one but a collection of NLG evaluators and not one but a collection of decision making settings, as well as to marginalise over choices and sources of stochasticity in evaluation. The findings suggest more robust conclusions are possible with the proposed approach.

**Strengths:**

1. Clear paper, except for cluttered notation here and there, but still mostly clear to me
2. Evaluation of UQ is important, the paper offers a critical take on the limitations of the typical UQ evaluation protocols, with a reasonable proposal for improvement
3. Proposed approach is simple, automated, appears to be effective

**Weaknesses:**

1. I think the paper needs an “oracle” experiment, where the errors in automated NLG evaluation are approximately eliminated by means of human eval. Of course, I don’t expect it to be as large scale as the rest, but still. Finding that the proposed approach allows us to get closer to the quality of conclusions derivable from this oracle setting is, to me, essential. With that in place, I can then take the larger scale, but more indirect, evidence presented with more optimism /
confidence that the seemly more coherent conclusions are indeed more meaningful.

That is, to me, the main weakness (unless I misunderstand something, but then I’m happy to be corrected) and the reason for my cautious  scores for soundness and contribution.

**Questions:**

Could you please address the weakness point above? Did I miss something in my interpretation of your results?

---

> ### Author Response · Authors · 2025-11-21
>
> We thank the reviewer for their appraisal of our submissions clarity as well as the simplicity and effectiveness of our approaches.
>
> With respect to the weakness pointed out by the reviewer:
>
> > I think the paper needs an “oracle” experiment, where the errors in automated NLG evaluation are approximately eliminated by means of human eval. Of course, I don’t expect it to be as large scale as the rest, but still. Finding that the proposed approach allows us to get closer to the quality of conclusions derivable from this oracle setting is, to me, essential. With that in place, I can then take the larger scale, but more indirect, evidence presented with more optimism / confidence that the seemly more coherent conclusions are indeed more meaningful.
>
> To provide such an oracle, we have manually labeled the correctness of 300 question-answer pairs on the CoQA produced by Llama 3 8B IT. We used 3 independent annotators with agreed upon labeling instructions in order to produce high quality annotations. The annotators were all volunteers driven purely by the desire to improve this paper. The resulting experiment description and correlation plot can be found in Appx.B.5.4. and Figure 7 of the updated manuscript. We find that MoJI has better agreement with human annotators than the individual judges do on average. Its agreement is often on par to that between human annotators. Intuitively, this makes perfect sense, as it is known that ensembling (which is what MoJI does in essence) is beneficial to the predictive quality.
>
> Currently, we placed this additional result in the appendix. However, we think about moving it to the main paper for the camera ready version, if the reviewer agrees to that.
>
> We hope this additional experiment increases the reviewers confidence that the more coherent conclusions are also more meaningful and leads to a positive reassessment of our work.

---

> > ### Comment · Reviewer_4ZXg · 2025-11-27
> >
> > Thank you for following up on this. It would be nice to move those to the main paper in a final version.
> >
> > To me, this clears my concerns with the paper (as I hope my original review already suggested). I’ve revised the scores accordingly.

---

### Official Review · Reviewer_oJfd · 2025-10-31

**Soundness:** 3
**Presentation:** 3
**Contribution:** 3
**Rating:** 6
**Confidence:** 4

**Summary:**

This paper systematically analyzes the pitfalls in evaluating uncertainty estimation methods for natural language generation (NLG), especially in the context of hallucination/confabulation detection in LLMs. The authors demonstrate that commonly used approximate correctness functions (e.g., BLEU, ROUGE, LLM-as-a-judge) can lead to substantial disagreement and bias in the ranking of uncertainty estimation methods. They propose using multiple alternative risk indicators, marginalizing over LLM-as-a-judge variants, and structured/perturbation tasks to improve robustness. The paper also introduces an Elo rating system for summarizing method performance across settings.

**Strengths:**

⦁	Accurate problem identification, experiments reveal key pitfalls in the evaluation field.
⦁	Evaluation suggestions have practical guidance value.
⦁	The paper is rigorous and well-argued.

**Weaknesses:**

⦁	Mainly focuses on evaluation methods themselves, with limited guidance for designing new uncertainty estimation methods.
⦁	Lacks ablation and robustness analysis for some suggestions (e.g., Elo rating, integrated evaluation).
⦁	Lacks deeper theoretical discussion on how to fundamentally eliminate evaluation noise and improve consistency.

**Questions:**

⦁	How stable is the Elo rating system across different datasets and evaluation metrics?
⦁	What is the computational cost and scalability of multi-metric fusion evaluation in large-scale practical evaluation?
⦁	Are there ablation experiments for different evaluation suggestions?
⦁	Are the paper's suggestions equally applicable to new NLG tasks (e.g., multi-turn dialogue, generative reasoning)?

---

> ### Author Response · Authors · 2025-11-21
>
> We thank the reviewer for their thoughtful assessment of our work. We are encouraged that our evaluation suggestions have been found to have practical guidance value and that our work was conducted rigorously and is well argued. Regarding the stated weaknesses and questions:
>
> > Mainly focuses on evaluation methods themselves, with limited guidance for designing new uncertainty estimation methods.
>
> We agree that our work only indirectly aims to advance uncertainty estimation methods by providing the means for robust evaluation. However, given the recent proliferation of methods published on this topic - evaluated under evidently frail protocols - our aim was to provide the foundation to reliably assess future advances in this important field.
>
> > Lacks deeper theoretical discussion on how to fundamentally eliminate evaluation noise and improve consistency.
>
> We would like to point out, that our approach of marginalizing the variability of approximate correctness (line 308ff) is motivated by the theoretical considerations on the effects of bias and variance in correctness labels on AUROC presented in lines 202 - 254 (more detailed theoretical discussion in Appendix D.2). As evaluation in ML is generally based on sampling, noise can not be fully eliminated, but our suggestions provide a tangible way to improve consistency.
>
> > How stable is the Elo rating system across different datasets and evaluation metrics?
>
> The Elo system does not inherently depend on datasets and metrics. From its point of view, there is only a set of pairwise experiments where one of the two “players” comes out on top.
> We did conduct an ablation detailed in responses to reviewers Fa32 and 3yHN (details in Appx.B.5.5., Fig.8 in the updated manuscript) which shows that in a partial overlap scenario, Elo score is better at retaining the ordering of methods than rank average.
>
> > What is the computational cost and scalability of multi-metric fusion evaluation in large-scale practical evaluation?
>
> This depends on the cost of individual metrics. The costliest ones are LLM-as-a-judge calls, yet our results in Figure 2 show that 4 calls already halve the variance of results and that there are diminishing returns past 10 judges. Devising a cost-optimal scheme could be a subject for future work.
>
> > Are there ablation experiments for different evaluation suggestions?
>
> We are unsure which type of ablation the reviewer is suggesting. The proposed evaluation strategies in Section 4 are orthogonal to each other, either suggesting alternative risk indicators or increasing the robustness of the selective prediction indicator. Using the ELO rating as aggregation (Section 5) is then done in a next step, after selecting reliable risk indicators for experiments, and aims to reliably and interpretably aggregate over a large set of experiments for comparing different uncertainty estimation methods.
>
> > Are the paper's suggestions equally applicable to new NLG tasks (e.g., multi-turn dialogue, generative reasoning)?
>
> This depends on the tasks. For example, mathematical reasoning tasks could be viewed as those with exact correctness. At the same time, symbolic answer extraction is a prominent limitation (e.g. Chandak et al. Incorrect Baseline Evaluations Call into Question Recent LLM-RL Claims | Notion) even though the answer space is much more restricted compared to the general QA.
> Multi-turn dialogue evaluation would be an interesting direction for future work that we didn’t consider before due to added complexity. Marginalizing over the variability of approximate correctness functions could directly be applicable to those settings. Overall we view the introduction of alternative useful evaluation settings for uncertainty estimation in NLG as a positive thing for the community.
>
> We look forward to the remaining discussion phase and hope that our rebuttal resolves the reviewer’s concerns.

---

### Official Review · Reviewer_3yHN · 2025-11-01

**Soundness:** 3
**Presentation:** 2
**Contribution:** 2
**Rating:** 8
**Confidence:** 4

**Summary:**

The paper argues that QA style selective prediction is brittle because approximate correctness functions may disagree and can be hacked. The proposal consists of alternative risk indicators (structured tasks with exact correctness, OOD, perturbation), SP-MoJI (marginalize across multiple judges/prompts/models), and Elo aggregation across settings.

The main motivation is that published research in the area mainly evaluates on QA with approximate correctness functions (Table 1), often with ROUGE/BLEU/BERTScore/LLM-as-judge, and little human evaluation. UE is formalized as ranking correlation between uncertainty and a risk indicator (AUROC of rank correlation). Three empirical properties that are posited as desirable are proposed, motivating three experiment families: SP (selective prediction), OOD detection, and perturbation detection. Correctness c_theta is defined and two label-perturbation effects on AUROC are shown. It is also shown how a sample-dependent bias yields a decomposition that mixes distorted/undistorted subsets, implying rank instabilities when we fail to marginalize. Figure 1 shows large disagreement among ROUGE/BLEU vs judge metrics, driven partly by extremely short reference answers; a ROUGE-2/BLEU implementation artifact is also reported. These disagreements translate into inconsistent UE method rankings. It is shown that correctness-hacking (Table 2) can substantially alter top-3 membership.

Following the analysis, the following remedies are proposed. a) Exact correctness via structured tasks (code unit tests, constrained generation), which avoids parameterized metrics. b) SP-MoJI: averages the outer correlation across multiple judges/prompts/models to marginalize judge aleatoric/epistemic uncertainty (Eq. 6). Further, bootstraps show a single judge gives SD≈0.04 in AUROC; ~4 judges halves SD; diminishing returns past ~10. c) OOD/perturbation: treats OOD identifiers or corruption strength as risk indicators, implements with Known-Unknowns, SQuADv2, and word-shuffle perturbations. In view of all this they use Elo to aggregate pairwise wins across (dataset x model x task) experiments; 400 Elo roughly corresponds to 1:10 odds. This is good for incomplete overlap across method evaluations and for subsets (QA vs code vs constrained text; instruction-tuned vs pre-trained; OOD vs perturbation).

**Strengths:**

The paper doesn't propose brand new math or concepts, but it does present a creative integration of fixes: outer-expectation marginalization, structured tasks as risk indicators, and Elo aggregation. All together, this can meaningfully update evaluation practice. The proposed fixes are easily actionable.

The analysis before the remedies are proposed (e.g. for AUROC under noise/bias, careful empirical demos for disagreement matrices, correctness hacking) is quite useful.

The paper is generally clearly written. Explanations are generally clear.

**Weaknesses:**

On page 5, the adversarial metric selection space does not seem to be pre-registered. Without a pre-defined grid, the "correctness-hacking" claim could itself involve cherry-picking.

One page 6, the SP-MoJI diversity factors seem under-specified. Which diversity (model family vs prompts vs decoding) contributes most to variance reduction?

On page 6: Multiple judges may still share family biases; cross-family calibration isn’t deeply analyzed. Wondering what the authors think of the potential impact.

On page 7: The paper is light on the Elo details. There is no K-factor/tuning, cycle handling, or uncertainty intervals; alternative ranking models not compared.

I also feel like the paper under-acknowledges the breadth of prior practice. slightly overstating QA dominance.

Finally, I also generally found the engagement with the broader literature slightly frustrating. The paper seems to cite papers by 2-3 groups (other than classical references), and develops around that. I am not going to suggest papers to cite, but placing the paper better would improve it.

**Questions:**

See the above.

---

> ### Author Response · Authors · 2025-11-21
>
> We thank the reviewer for their in-depth assessment of our work. We are pleased that our proposed fixes to the evaluation procedure are found to be meaningful, creative and easily actionable.
>
> To address the stated weaknesses:
>
> > On page 5, the adversarial metric selection space does not seem to be pre-registered. Without a pre-defined grid, the "correctness-hacking" claim could itself involve cherry-picking.
>
> This is a very good point. Indeed, we have only provided a brief verbal summary of the grid used for the Table 2.
> We have modified the manuscript to feature a full description of correctness functions used for the correctness hacking table in the appendix and provided a reference in the caption of Table 2.
> The range of correctness functions and thresholds was relatively conservative: Judge-Llama3-70B, Rouge-L, Rouge-L-Sum with threshold options of [0.3,0.5]. Those choices were commonly encountered in prior work. With an even larger set of correctness functions, e.g. addition of bleu or bert based metrics, the problem of correctness hacking could likely be further exacerbated.
>
> > One page 6, the SP-MoJI diversity factors seem under-specified. Which diversity (model family vs prompts vs decoding) contributes most to variance reduction?
>
> This is indeed an interesting question and answering it would make a good addition to the paper. To that end we currently rerun the evaluation for Figure 2 with an even larger variety of judges/prompts and sampling iterations. We will try to provide additional results for that in subsequent comments later in the rebuttal, since this will require more time.
>
> > On page 6: Multiple judges may still share family biases; cross-family calibration isn’t deeply analyzed. Wondering what the authors think of the potential impact.
>
> The impact of model family biases is there. For example, they could react differently to instruction prompts or put different linguistic emphasis when dealing with ambiguous answers. We observe in correlation plots that Qwen and Llama models often correlate less between the families compared to within the family. More detailed investigation of these phenomena could be a subject for future work.
>
> > On page 7: The paper is light on the Elo details. There is no K-factor/tuning, cycle handling, or uncertainty intervals; alternative ranking models not compared.
>
> We thank the reviewer for raising this interesting point.
>
> With respect to Elo computation details: we stuck to the most commonly used settings. Detailed description is provided in B.5.2. The only setting we changed was K (usually set to 16 or 32, we set it to 2). This setting does not affect the values to which procedure converges, but rather the speed of convergence. The small values of K are generally considered to be a safe bet, but impose some requirements on the number of games. We sample ‘games’ with replacement 100,000 times and achieve convergence to stationary distribution (Figure 6). Wallclock-wise it still requires ~10 seconds.
>
> As for the comparison and ablations on Elo score, we have added a comparison to average rank under incomplete overlap (Appx.B.5.5. And Figure 8 in the revised manuscript). In this ablation we compare the abilities of the rank average and Elo score to retain order under occlusion of parts of the experimental suite. Elo score shows better retention under such conditions.
> If the reviewer has any additional ablation scenarios in mind, we would gladly perform them and incorporate the results into our work.
>
> > I also feel like the paper under-acknowledges the breadth of prior practice. slightly overstating QA dominance.
>
> If the reviewer has any particular wording in mind, we would gladly soften the tone. Indeed, we acknowledge that over-reliance on QA tasks somewhat exacerbated in recent works; the evaluation protocols of more classical works were broader.
>
> > Finally, I also generally found the engagement with the broader literature slightly frustrating. The paper seems to cite papers by 2-3 groups (other than classical references), and develops around that. I am not going to suggest papers to cite, but placing the paper better would improve it.
>
> We tried to be very inclusive of different works, but this field expanded very fast in the last 2-3 years. For example, the works cited in Table 1 originate from 9 groups without author overlap, likely being disjoint groups as far as we can tell without checking the authors individual affiliation histories. We selected those due to being published at top venues and being influential in many of the subsequent publications on the topic we read. If the reviewer gives any direction of classical references that we should include, we would happily do so to further improve the exposition of our work.
>
> We thank the reviewer again for their thoughtful feedback and look forward to the remaining rebuttal.

---

> > ### Comment · Reviewer_3yHN · 2025-11-27
> >
> > Thank you! I went through the responses in general, and would keep my rating (which is a clear accept). I hope the authors are able to incorporate all the promised changes (other than those not already added e.g. B 5.3-5.5)..

---

### Official Review · Reviewer_Fa32 · 2025-11-04

**Soundness:** 4
**Presentation:** 3
**Contribution:** 2
**Rating:** 6
**Confidence:** 4

**Summary:**

The paper highlights the problem that unreliable generation quality estimation metrics can further mislead the uncertainty estimation evaluation. The authors demonstrate that you can hack the evaluation metric by choosing the proper LLM generation quality metric. This problem is not unique for uncertainty, but it is a challenge for the LLM evaluation in general.

The authors suggest two methods:
1.	They suggest using multiple LLMs as a judge to reduce variance of generation quality assessment.
2.	They propose using ELO rating for aggregating the scores across multiple different tasks and datasets.
Two problems with the paper:
1.	Bias due to inadequate choice of quality metrics were investigated previously in (Santilli et al., 2025) and some solutions to this problem were suggested in (Santilli et al., 2025) and (Vashurin et al., 2025).
2.	ELO rating looks redundant compared to simple averaging of metrics. It would be great if you could clarify the situations when it is better than simple averaging.  I think the paper could benefit from better analysis of ELO rating (cases when it is really needed).

Literature:
Andrea Santilli, Adam Golinski, Michael Kirchhof, Federico Danieli, Arno Blaas, Miao Xiong, Luca Zappella, and Sinead Williamson. 2025. Revisiting Uncertainty Quantification Evaluation in Language Models: Spurious Interactions with Response Length Bias Results. In Proceedings of the 63rd Annual Meeting of the Association for Computational Linguistics (Volume 2: Short Papers), pages 743–759, Vienna, Austria. Association for Computational Linguistics. https://aclanthology.org/2025.acl-short.60/

**Strengths:**

The authors suggest two methods:
1.	They suggest using multiple LLMs as a judge to reduce variance of generation quality assessment.
2.	They propose using ELO rating for aggregating the scores across multiple different tasks and datasets.

**Weaknesses:**

Two problems with the paper:
1.	Bias due to inadequate choice of quality metrics were investigated previously in (Santilli et al., 2025) and some suggestions to this problem were suggested in (Santilli et al., 2025) and (Vashurin et al., 2025).
2.	ELO rating looks redundant compared to simple averaging of metrics. It would be great if you could clarify the situations when it is better than simple averaging.  I think the paper could benefit from better analysis of ELO rating (cases when it is really needed).

**Questions:**

Can you provide particular examples that motivate ELO?

---

> ### Author Response · Authors · 2025-11-21
>
> We thank the reviewer for their positive assessment of our work and concise questions.
>
> We would like to address the weaknesses as follows:
>
>
> 1. We thank the reviewer for bringing up these two works. We are well aware of them and cite and discuss each throughout the paper. We do our best to clearly state where our insights and proposed solutions to the identified problems go beyond prior work. Are there any particular details on the contributions of these prior works that we should discuss in more depth in our paper?
> 2. (and the question) We thank the reviewer for bringing up this important point. While we do speculate about several scenarios where the Elo score would have advantages, we do not provide any experiments to back these claims. To address this, we have created a synthetic scenario that is described in Appx.B.5.5 of the updated manuscript. Briefly: it consists of removing some of the entries from our experimental set to create the indirect comparison (low overlap) situation. We then compare how well does the Rank Average and Elo score retain the global method ordering compared to the unmodified full experimental suite. We show (Figure 8 of the updated manuscript) that Elo is noticeably better at preserving the ordering than rank averaging (Kendall tau of 0.714 vs 0.600 on the top half of the methods).
>
> The changes in the revised manuscript are highlighted in purple and teal corresponding to text addition and deletion accordingly in order to highlight the differences.
>
> We are looking forward to the reviewers further feedback. We are open to additional proposals on ablation of Elo score aggregation as well as to specific suggestions on more detailed attribution of the two prior works mentioned.

---

### Comment · Area_Chair_7mWx · 2025-11-23

Dear Reviewers,

The authors have submitted their rebuttal addressing your reviews. Please take the time to:

1. Read the rebuttal carefully
2. Ask clarifying questions if anything remains unclear
3. Update your scores and reviews based on the authors' responses

Please be mindful of timing: If you have follow-up questions for the authors, **post them early enough to give them adequate time to respond** before the discussion period closes on December 3rd.

Your timely engagement is crucial for a fair and thorough review process.

Thank you for your continued effort on this paper.

Best regards,
Area Chair

---

### Author Response · Authors · 2025-12-01
**General Response**

We thank the reviewers for their thoughtful and largely positive feedback. The main changes to the paper in the process of the rebuttal can be summarized as follows:

1. **Benefits of Elo score and ablations.**
Requested by reviewers Fa32, 3yHN and oJfd. We have added an ablation for the indirect comparison to the paper (Appx. B.5.5. / Fig.8) where we compare the average rank to Elo scores ability to retain ordering under occlusion of part of the experimental suite.
2. **Human Annotation Correlation experiment.**
Requested as the main point by reviewer 4ZXg. We have manually labeled 300 examples for the CoQA task and checked the correlation of MoJI to Humans. MoJI shows higher correlation to human annotators than individual judges on average. This was the reviewers sole concern and it was addressed to satisfaction, *prompting them to increase their score from 4 to 8 before the scores got reverted.*

The reviewers largely appraised the paper's clarity and actionability of the proposed methodology. The changes that were necessary to address the concerns of the reviewers were relatively minor, consisting of the auxiliary experiments described above. Reviewers who managed to leave further feedback before the abrupt end of the discussion period (3yHN and 4ZXg) were satisfied with the additional experiments and answers provided as part of the rebuttal. At this point, there are no reviewers who are in opposition to this paper's acceptance.

---

### Meta-Review · Area_Chair_a9kj · 2026-01-06

**Summary:**

The paper highlights issues with existing uncertainty quantification metrics for natural language generation, for instance, it shows how existing uncertainty metrics can be misleading. The work suggests using multiple LLM-as-a-judge as an ensemble and shows that this can reduce evaluation biases, and a new ELO rating of uncertainty estimation methods is introduced.

Overall, the paper received high scores. 4ZXg was planning to change scores from 4 to 8, making the average score of this paper to be 7. Reviewers main concerns about similarities to previous works, soundness of the ELO rating and need for an "oracle experiment" using human evals were addressed in rebuttals and comments to the AC.

As all reviewers liked the contribution, and I dont see any issues with it, the paper is recommended for acceptance.

**Reviewer Concerns:**

> Fa32 raised concerns about similarities to previous works (Santilli et al., 2025) and (Vashurin et al., 2025)
* These works can be seen as concurrent work, and they cite an earlier workshop version of this paper.

> Issues or benefit of the newly introduced ELO rating (Fa32, 3yHN, oJfd)
* Addressed by additional experiments and ablations in the paper

> Need for "oracle experiment" via human evals
* Added to the appendix

**Reviewer Scores:**

Reviewers would have likely maintained the scores, except for 4ZXg who indicated a score change from 4 to 8, giving an average rating of 7 for this paper.

---

### Decision · Program_Chairs · 2026-01-26

Accept (Poster)